# Neoadjuvant relatlimab and nivolumab in resectable melanoma

Rodabe N. Amaria[1,12 ✉], Michael Postow[2,12], Elizabeth M. Burton[1,12], Michael T. Tetzlaff[3], Merrick I. Ross[4], Carlos Torres-Cabala[5], Isabella C. Glitza[1], Fei Duan[6], Denái R. Milton[7], Klaus Busam[8], Lauren Simpson[1], Jennifer L. McQuade[1], Michael K. Wong[1], Jeffrey E. Gershenwald[4], Jeffrey E. Lee[4], Ryan P. Goepfert[9], Emily Z. Keung[4], Sarah B. Fisher[4], Allison Betof-Warner[2], Alexander N. Shoushtari[2], Margaret Callahan[2], Daniel Coit[10], Edmund K. Bartlett[10], Danielle Bello[10], Parisa Momtaz[2], Courtney Nicholas[6], Aidi Gu[6], Xuejun Zhang[6], Brinda Rao Korivi[11], Madhavi Patnana[11], Sapna P. Patel[1], Adi Diab[1], Anthony Lucci[4], Victor G. Prieto[5], Michael A. Davies[1], James P. Allison[6], Padmanee Sharma[6,13], Jennifer A. Wargo[4,13], Charlotte Ariyan[10,13] & Hussein A. Tawbi[1,13]

Relatlimab and nivolumab combination immunotherapy improves progression-free survival over nivolumab monotherapy in patients with unresectable advanced melanoma[1]. We investigated this regimen in patients with resectable clinical stage III or oligometastatic stage IV melanoma (NCT02519322). Patients received two neoadjuvant doses (nivolumab 480 mg and relatlimab 160 mg intravenously every 4 weeks) followed by surgery, and then ten doses of adjuvant combination therapy. The primary end point was pathologic complete response (pCR) rate[2]. The combination resulted in 57% pCR rate and 70% overall pathologic response rate among 30 patients treated. The radiographic response rate using Response Evaluation Criteria in Solid Tumors 1.1 was 57%. No grade 3–4 immune-related adverse events were observed in the neoadjuvant setting. The 1- and 2-year recurrence-free survival rate was 100% and 92% for patients with any pathologic response, compared to 88% and 55% for patients who did not have a pathologic response (P = 0.005). Increased immune cell infiltration at baseline, and decrease in M2 macrophages during treatment, were associated with pathologic response. Our results indicate that neoadjuvant relatlimab and nivolumab induces a high pCR rate. Safety during neoadjuvant therapy is favourable compared to other combination immunotherapy regimens. These data, in combination with the results of the RELATIVITY-047 trial[1], provide further confirmation of the efficacy and safety of this new immunotherapy regimen.

Patients with locoregionally advanced, resectable melanoma have a high risk of relapse and death from melanoma[3]. Specifically, patients with clinically detected nodal disease have a risk of melanoma-specific mortality that could be as high as 75%[3]. Although current adjuvant therapy decreases the risk of recurrence by about 50% (BRAF-targeted therapy hazard ratio (HR) 0.49, single agent PD-1 HR approximately 0.54)[4,5], there has yet to be confirmation of the impact on overall survival[4,6]. In an attempt to intensify therapy beyond single agent anti-PD-1, the Checkmate-915 trial was designed to investigate if the addition of ipilimumab to nivolumab in the adjuvant setting improved recurrence-free survival (RFS) compared to nivolumab alone. The combination of ipilimumab and nivolumab did not improve RFS (HR 0.92) and it significantly increased toxicity (grade 3–4 adverse events (AEs) 43%, compared to 23% for single agent anti-PD-1)[7], indicating that intensification of adjuvant therapy with ipilimumab and nivolumab in the adjuvant setting is not the optimal approach for improving recurrence outcomes.

Neoadjuvant therapy offers several advantages over upfront surgery and adjuvant therapy, including potential for improvement in clinical outcomes and understanding molecular and immunological mechanisms of treatment response and resistance[8–13]. Additionally, neoadjuvant immunotherapy has demonstrated ability in preclinical models and in human samples to increase expansion of antigen-specific T cells due to the presence of tumour at the time of treatment compared to

[1]Department of Melanoma Medical Oncology, The University of Texas MD Anderson Cancer Center, Houston, TX, USA. [2]Department of Medicine, Memorial Sloan Kettering Cancer Center and Weill Cornell Medical College, New York, NY, USA. [3]Department of Pathology, The University of California San Francisco, San Francisco, CA, USA. [4]Department of Surgical Oncology, The University of Texas MD Anderson Cancer Center, Houston, TX, US. [5]Department of Pathology, The University of Texas MD Anderson Cancer Center, Houston, TX, USA. [6]Department of Immunology, The University of Texas MD Anderson Cancer Center, Houston, TX, USA. [7]Department of Biostatistics, The University of Texas MD Anderson Cancer Center, Houston, TX, USA. [8]Department of Pathology, Memorial Sloan Kettering Cancer Center, New York, NY, USA. [9]Department of Head and Neck Surgery, The University of Texas MD Anderson Cancer Center, Houston, TX, USA. [10]Department of Surgical Oncology, Memorial Sloan Kettering Cancer Center, New York, NY, USA. [11]Department of Radiology, The University of Texas MD Anderson Cancer Center, Houston, TX, USA. [12]These authors contributed equally: Rodabe N. Amaria, Michael Postow, Elizabeth M. Burton. [13]These authors jointly supervised this work: Padmanee Sharma, Jennifer A. Wargo, Charlotte Ariyan, Hussein A. Tawbi. ✉e-mail: rnamaria@mdanderson.org

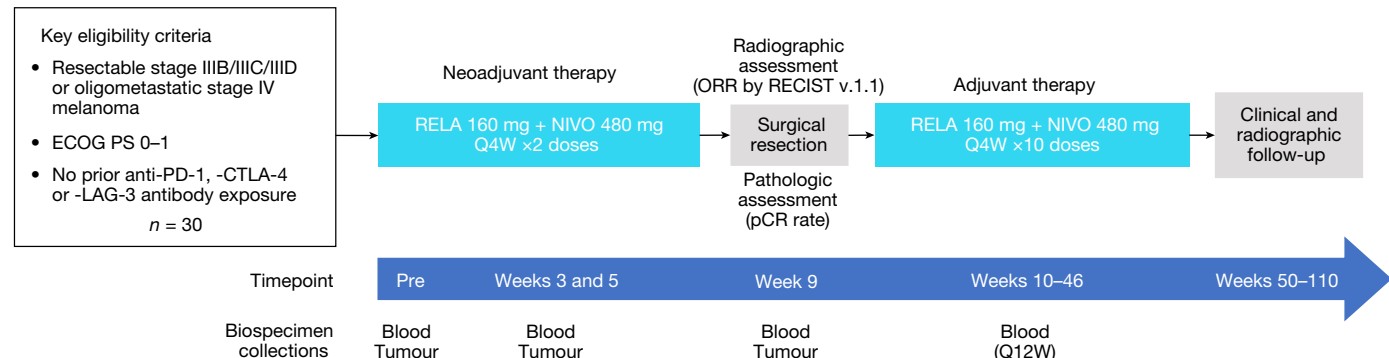

**Fig. 1 | Study design.** Eligible patients receive two doses of relatlimab 160 mg with nivolumab 480 mg intravenously every 4 weeks (Q4W) in the neoadjuvant setting and then have repeat imaging for calculation of RECIST response. Surgery takes place at week 9 for evaluation of pathologic response. Patients receive up to ten doses of relatlimab 160 mg and nivolumab 480 mg every 4 weeks in the adjuvant setting and are followed for 2 years for evidence of recurrence. Blood and tumour are collected during screening, at weeks 3, 5 and at time of surgery at week 9. Blood is collected every 12 weeks (Q12W) in the adjuvant setting. ECOG PS, Eastern Cooperative Oncology Group Performance Status; RELA, relatlimab; NIVO, nivolumab; ORR, objective response rate; RECIST, Response Evaluation Criteria in Solid Tumors.

the expansion seen when the same immunotherapy is administered in the adjuvant setting[14,15]. The neoadjuvant setting also offers the opportunity to intensify therapy with combinations for a short pre-operative course, allowing for a direct estimate of therapeutic efficacy and the ability to inform adjuvant therapy decisions.

One potential limitation of neoadjuvant immunotherapy is delay in curative-intent surgery if grade 3/4 immune-related adverse events (IRAEs) occur during treatment. For example, neoadjuvant administration of 2–3 doses of ipilimumab 3 mg kg⁻¹ + nivolumab 1 mg kg⁻¹ was associated with 73–90% grade 3/4 toxicities, which led to surgical delays in approximately 27% of patients[15,16]. The OpACIN-NEO trial compared two doses of neoadjuvant therapy with different dosing strategies of ipilimumab and nivolumab. This study demonstrated that ipilimumab 1 mg kg⁻¹ with nivolumab 3 mg kg⁻¹ showed an at least equivalent pCR rate (57%) to the ipilimumab 3 mg kg⁻¹ + nivolumab 1 mg kg⁻¹ regimen (47%), but with a lower (20% versus 40%) incidence of grade 3/4 toxicities[17]. These data highlight the goal of identifying new regimens that enhance pathologic responses and reduce risk of recurrence with improved toxicity profiles.

The lymphocyte-activation gene 3 (LAG-3) regulates an inhibitory immune checkpoint limiting T cell activity and is a marker for T cell exhaustion[18,19]. Relatlimab is a human IgG4 LAG-3-blocking monoclonal antibody that restores the effector function of exhausted T cells and has been investigated in both checkpoint inhibitor-naïve (NCT03470922)[1] and refractory metastatic melanoma (NCT01968109)[20]. In the randomized phase 2/3 RELATIVITY-047 study, the combination of relatlimab with nivolumab in patients with treatment-naïve unresectable stage III or stage IV metastatic melanoma demonstrated significant improvement in progression-free survival compared to single agent nivolumab (HR 0.78 (95% confidence interval (CI), 0.64–0.94)). Moreover, the combination was well tolerated with 21.1% of patients experiencing grade 3/4 treatment-related AEs[1]. Given its efficacy and favourable toxicity profile, this combination therapy received US Food and Drug Administration approval for use in patients with metastatic melanoma on 18 March 2022.

Our group previously published our experience of a randomized, investigator-initiated clinical trial of either single agent nivolumab (240 mg intravenously every 2 weeks up to four doses) or nivolumab 1 mg kg⁻¹ with ipilimumab 3 mg g⁻¹ (intravenously every 3 weeks up to three doses) in the neoadjuvant setting[16]. In this trial, we concluded that although neoadjuvant single agent nivolumab was safe (8% grade 3/4 toxicities), its efficacy was modest (25% pCR rate). Although the combination of nivolumab with ipilimumab was effective with a 45% pCR rate, the toxicity was prohibitively high with 73% grade 3/4 toxicities[16]. Given these data and the early closure of the study due to suboptimal performance of both treatment arms, our team sought to evaluate new immunotherapy combinations with the intention of preserving pathologic response while minimizing toxicities. We opened a new arm to this existing prospective clinical trial to determine pCR rate, safety and efficacy of the relatlimab and nivolumab combination in patients with resectable clinical stage III or oligometastatic stage IV melanoma (Clinicaltrial.gov number NCT02519322) (Fig. 1). Here we report the clinical results and immune profiling of this neoadjuvant therapy combination.

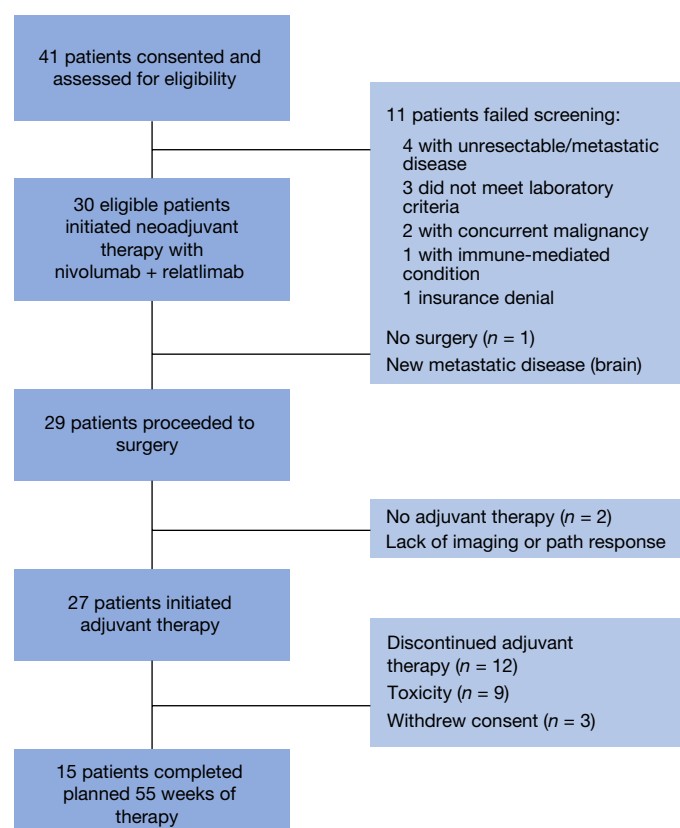

**Fig. 2 | Consort diagram and patient disposition.** A total of 41 patients were screened for protocol and there were 11 screen failures and 30 patients were eligible to initiate therapy. After completion of neoadjuvant therapy, one patient developed distant metastases and did not proceed to surgery. Twenty-nine patients proceeded to surgery and 17 patients (57%) achieved a pCR. Twenty-seven patients initiated adjuvant therapy and 15 went on to complete entire duration of treatment. path, pathologic.

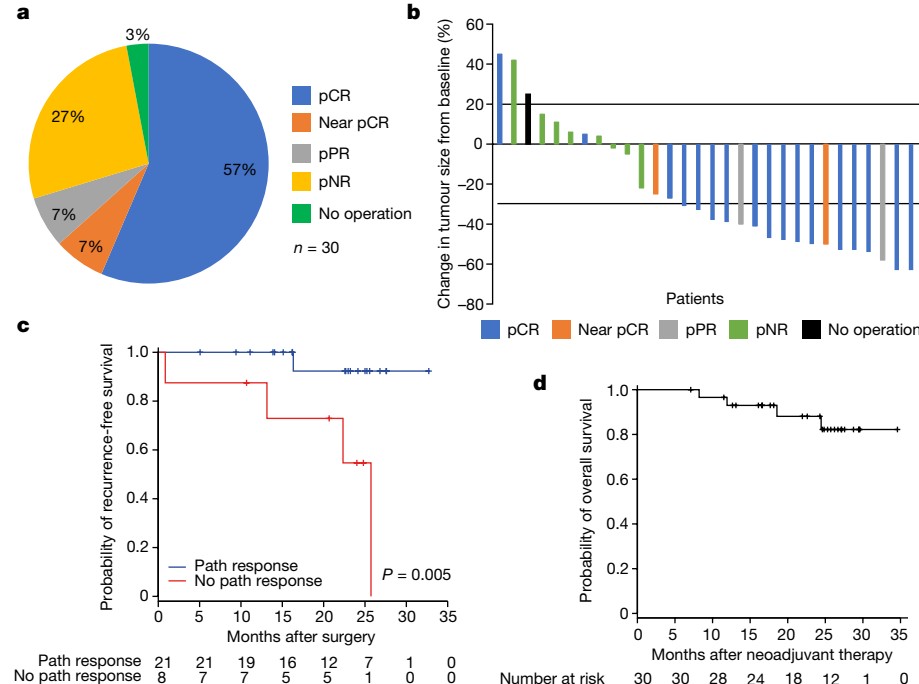

**Fig. 3 | Response data and long-term outcomes. a**, Breakdown of pathologic responses for the 29 patients who underwent surgery as interpreted by the guidelines of the INMC. Result details (values in chart rounded): no operation, 1 of 30 patients (3.33%); pCR, 17 of 30 patients (56.67%); near pCR, 2 of 30 patients (6.67%); pPR, 2 of 30 patients (6.67%); pNR, 8 of 30 patients (26.67%). **b**, Waterfall plot of neoadjuvant response as per RECIST 1.1 criteria with colour coding indicating pathologic response. pCR indicates lack of viable tumour. Near pCR indicates greater than 0% but less than or equal to 10% viable tumour, pPR is greater than 10% to less than or equal to 50% viable tumour and pNR is greater than 50% viable tumour. **c**, Probability of being relapse-free based on any pathologic response versus no pathologic response. **d**, Overall survival curves for the entire cohort.

## Patient characteristics

From 19 September 2018 to 23 September 2020, 41 patients were consented and 30 passed screening evaluations and were treated at MD Anderson Cancer Center and Memorial Sloan Kettering Cancer Center. The most common reasons for screen failure included lack of resectable disease as determined by multidisciplinary review ($n$ = 4 patients) and laboratory values outside the specified criteria ($n$ = 3 patients) (Fig. 2).

The median age of treated patients was 60 (range 35–79) and 63% of patients were male (Extended Data Table 1). Melanoma clinical stage was 60% stage IIIB, 26% IIIC, 7% IIID and 7% M1A by the American Joint Committee on Cancer 8th edition criteria[3]. Thirty-three per cent of patients had de novo clinical stage III or oligometastatic stage IV melanoma, and 67% had prior melanoma surgery. Only 17% of patients had *BRAF*-mutated melanoma, probably due to enrolment on a competing neoadjuvant trial specific for patients with *BRAF*-mutated disease. Only one patient had prior systemic therapy (BRAF and MEK inhibition). The median target lesion sum of diameters was 26 mm (Extended Data Table 1).

## Patient disposition

Of the 30 treated patients, 29 were able to receive the planned two doses of neoadjuvant relatlimab and nivolumab. One patient received only one dose due to asymptomatic troponin elevations with concern for myocarditis, which was eventually determined to not be attributable to neoadjuvant immunotherapy after the patient underwent myocardial biopsy and was able to proceed safely to surgery. One patient did not proceed to surgery due to development of distant metastatic disease during neoadjuvant therapy. Of the 29 patients that underwent surgery, 27 patients proceeded to surgery as scheduled at week 9; one patient was delayed due to the aforementioned myocarditis toxicity concern and one patient was delayed due to SARS-CoV2 pandemic-related hospital surgery restrictions. Twenty-seven patients proceeded with adjuvant

therapy and two patients elected to not proceed with adjuvant therapy due to suboptimal pathologic and imaging response. Fifty-six per cent of patients completed the entire duration of protocol therapy, 33% of patients discontinued adjuvant therapy due to toxicity and 11% of patients withdrew consent during adjuvant therapy (Fig. 2). Currently, all patients are off protocol therapy.

## Clinical activity

Of the 30 patients enroled, 29 patients underwent surgery (97%), 17 (57%; 95% CI, 37–75%) achieved pCR, two (7%) near pCR (defined as greater than 0% but less than or equal to 10% viable tumour), two (7%) partial pathologic response (pPR; defined as greater than 10% to less than or equal to 50% viable tumour) and eight (27%) no pathologic response (pNR; defined as greater than 50% viable tumour) (Fig. 3a). A major pathologic response (pCR + near pCR) was achieved in 63% of patients and any pathologic response (pCR + near pCR + pPR) in 70% of patients[2].

The radiographic overall response rate was 57% (all partial responses (PRs); 33% had stable disease (SD) and 10% had progressive disease (PD) (Fig. 3b)) in the intention-to-treat population. Pathologic response was frequently disconcordant with radiographic response at 8 weeks. For example, of the 19 patients who achieved major pathologic response (pCR and near pCR), one patient had radiographic PD, three had SD and 15 had PR. Of the eight patients with pNR, only one had radiographic PD and seven had SD. In the 16 patients with tumour sum of diameters at the median or higher (at least 26 mm), there was a mix of Response Evaluation Criteria in Solid Tumors (RECIST; 6% PD, 38% SD, 56% PR) and pathologic responses (38% pNR, 6% pPR, 6% near pCR, 50% pCR), indicating that baseline tumour burden did not correlate directly with pathologic or radiographic response.

With a median follow-up of 24.4 months (range 7.1–34.6 months) for the 30 treated patients, 1- and 2-year event-free survival rates (time

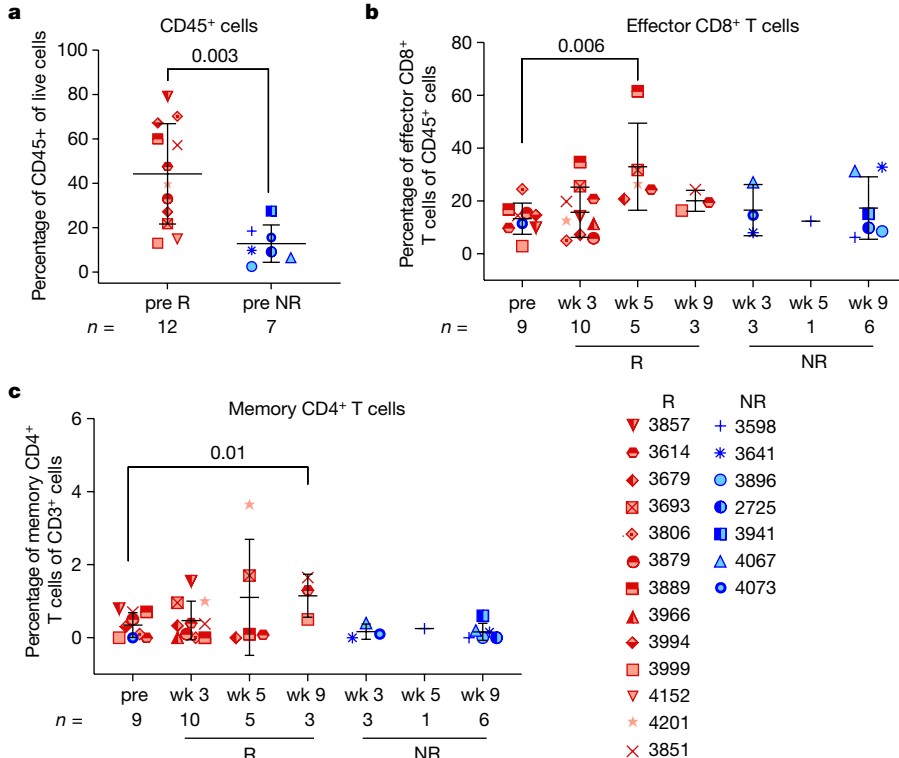

**Fig. 4 | Correlative analyses in tumour specimens.** Tumour tissue samples harvested from patients at baseline, and post relatlimab and nivolumab treatment were analysed in a single experiment by CyTOF (**a**–**c**). **a**, Frequency of CD45[+] cells was assessed through manual gating. **b**, Frequency of an effector CD8[+] T cell subset (CD3[+]CD8[+]CD45RO[low]) in unsupervised clustering is shown. **c**, Frequency of a memory CD4[+] subset (CD45RO[+]ICOS[+] TCF7[+]BTLA[+]CD28[+]TIGIT[+]) was determined by unsupervised clustering. Data shown in **a**–**c** are mean ± s.d., and *n* values are indicated in the figure. *P* values shown in each graph were calculated by two-tailed unpaired *t*-test, with no multiple comparisons. Red indicates pathologic responders; blue, non-responders. CyTOF, mass cytometry; NR, non-responder; R, responder; wk, week.

from treatment initiation to recurrence in all patients) were 90% and 81%, respectively (Extended Data Fig. 1). The 1- and 2-year RFS rates (time from surgery to recurrence in patients that underwent surgery) were 97% and 82%, respectively (Extended Data Fig. 2a). The 1- and 2-year RFS rates were 100% and 91% for patients with pCR, compared to 92% and 69% for those without pCR (*P* = 0.10) (Extended Data Fig. 2b). The 1- and 2-year RFS rates were 100% and 92% for patients with any pathologic response, compared to 88% and 55% for those without a pathologic response (*P* = 0.005) (Fig. 3c). The 1- and 2-year overall survival rates for all patients were 93% and 88% (Fig. 3d).

Of the three patients with RECIST PD to neoadjuvant therapy, one patient developed distant metastases (brain) and did not undergo surgery. The two other RECIST PD patients appeared to progress locally in the involved nodal basin only, and complete surgical resection was achieved for both. One of these patients did not proceed with adjuvant therapy due to pNR and patient/physician decision; the other achieved a pCR, proceeded with adjuvant therapy and completed protocol therapy without disease recurrence (Fig. 2). Two patients (both pNR) experienced local recurrence in soft tissue adjacent to site of prior surgical resection at 3 and 14 months after completion of all ten doses of adjuvant therapy. One patient with pCR reportedly experienced unconfirmed disease progression in the brain and passed away 14 months after surgery.

## Safety

There were no grade 3/4 IRAEs during the 8 weeks of neoadjuvant therapy (Extended Data Table 2). Twenty-six per cent of patients developed grade 3/4 IRAEs in the adjuvant setting (from week 9 and beyond) (Extended Data Table 2). Overall, 33% of patients elected to discontinue adjuvant

therapy due to any toxicity (most commonly transaminitis). Although there were asymptomatic troponin elevations, no patients experienced symptomatic troponin elevations, myocarditis or other cardiac toxicity attributable to study medications as assessed by cardiology consultation. The most frequent IRAE was secondary adrenal insufficiency (23%), with none of the patients experiencing adrenal recovery to date.

## Correlative studies

Biomarker analysis focused on characterizing immune cell subsets in the tumour microenvironment and peripheral blood was performed by mass cytometry (CyTOF) and flow cytometry. LAG-3 and PD-1 levels in baseline tumour samples did not correlate with pathologic response (Extended Data Fig. 3). In tumours, the frequency of CD45[+] cells was higher in pretreatment samples of responders, defined as patients with less than 50% tumour viability at surgery, compared to pretreatment samples of non-responders (NRs; greater than or equal to 50% tumour viability) (Fig. 4a) by CyTOF. Unsupervised clustering identified an effector CD8[+] T cell subset (CD8[+]CD45RO[low]) and a memory CD4[+] T cell subset (CD4[+]CD45RO[+]TCF7[+]CD28[+]BTLA[+]TIGIT[+]) that were increased in posttreatment tumour specimens versus pretreatment in patients with favourable response (Fig. 4b,c). The increases in these cell populations were not appreciated in the NR patient group, although it should be noted that the number of evaluable specimens was low in this group (Fig. 4b,c). By contrast, the frequency of an M2-like macrophage subset decreased in tumours after treatment in patients with favourable response (Extended Data Fig. 4a). In blood, there was a trend for increased EOMES[+]CD8[+] T cells in patients with favourable versus non-favourable response after treatment, with largest differences seen at week 5 posttreatment (Extended Data Fig. 4b).

## Discussion

In patients with resectable clinical stage III or oligometastatic stage IV melanoma, neoadjuvant relatlimab with nivolumab resulted in high pCR rate (57%; 95% CI, 37–75%) and improvement in the 2-year RFS rate in patients who achieved any pathologic response compared to those without a pathologic response (P = 0.005). The lower limit CI (37%) exceeded the minimum target of 30% in the study design. This regimen was tolerated well in the neoadjuvant setting, with 26% grade 3 toxicities noted with continued dosing in the adjuvant setting. In patients with pathologic response, increased immune cell infiltration was identified at baseline and decreased M2 macrophages were demonstrated over the course of neoadjuvant therapy.

The first two randomized arms of this trial evaluated both single agent nivolumab and the combination of ipilimumab 3 mg kg$^{-1}$ and nivolumab 1 mg kg$^{-1}$. Twenty-seven per cent of patients treated with ipilimumab 3 mg kg$^{-1}$ and nivolumab 1 mg kg$^{-1}$ required surgical delays of 1–10 weeks due to need for steroids and prolonged steroid taper[16]. With no grade 3/4 IRAEs observed in the neoadjuvant setting and no confirmed toxicity-related surgical delays, the combination of nivolumab and relatlimab now provides complementary information and demonstrates a highly effective regimen with manageable toxicities in the neoadjuvant setting.

Although there were no grade 3/4 IRAEs in the neoadjuvant setting, 26% grade 3/4 toxicities were experienced in the adjuvant setting. The most common IRAE observed was secondary adrenal insufficiency. As 33% of patients discontinued therapy before the planned full year of treatment, due to toxicity, it raises questions of whether continued dosing in the adjuvant setting is necessary following pathologic response to neoadjuvant therapy. Additionally, none of the patients who stopped therapy early due to toxicity have experienced a recurrence event. There is not clear consensus on the need for the adjuvant phase of therapy within neoadjuvant trials, with completed or ongoing trials including complete omission of any adjuvant therapy, use of adjuvant therapy only in poor responders or adjuvant therapy to complete 1 year of treatment[8,15–17,21–23]. Additionally, the use of adjuvant therapy can certainly affect the RFS and can cloud the interpretation of neoadjuvant therapy data. Understanding the contribution of adjuvant immunotherapy following immunotherapy in the neoadjuvant setting to clinical benefit remains an active area of research interest.

The historic dogma in neoadjuvant chemotherapy emphasized pCR as the critical end point correlating with the most durable clinical outcomes[11–13]. This was similarly appreciated in the International Neoadjuvant Melanoma Consortium (INMC) pooled analysis of neoadjuvant BRAF/MEK inhibitor use in patients with clinical stage III melanoma, showing that achieving a pCR, but not a pPR, correlated with improved RFS[9,16,17,21]. Although the pCR end point may still be appropriate for neoadjuvant chemotherapy or molecularly targeted therapy, our data provide further evidence that in the context of neoadjuvant immunotherapy in melanoma, any pathologic response (less than 50% viable tumour) is associated with favourable long-term clinical outcomes (Fig. 3c)[9,16,17,21]. Similar patterns of improved clinical responses with any pathologic response are being appreciated in neoadjuvant immunotherapy trials across solid tumours[24–26].

Although baseline LAG-3 and PD-1 levels in tumour samples did not correlate with response, we observed increased frequencies of memory CD4$^+$ and effector CD8$^+$ T cells in the posttreatment tumour specimens of patients with favourable treatment response. These findings are concordant with previous studies in which responses to anti-PD-1 were associated with higher CD8$^+$ T cells[15–17,21,27,28]. Furthermore, we observed a reduction in M2-like macrophages with treatment only in the patients that achieved a pathologic response, possibly serving as a target to further improve responsiveness to this regimen, and/or to further evaluate in other studies of nivolumab plus relatlimab[29]. Analysis of longitudinal peripheral blood specimens by flow cytometry revealed higher frequency of EOMES$^+$CD8$^+$ T cells in posttreatment samples of responding patients, suggesting CD8$^+$ T cells expressing EOMES could contribute to tumour regression. This supports a potentially critical role of EOMES for antitumour activity of CD8$^+$ T cells, as previously described[30]. These data indicate that a higher frequency of total immune cell infiltration, as well as increased specific effector CD4$^+$ and CD8$^+$ T cell subsets, with a concomitant decrease in suppressive myeloid cells in the tumour microenvironment, correlate with clinical response to this regimen in the neoadjuvant setting. It should be noted that the number of usable samples in the NR patients was low, which limits comparative correlative analyses in this study.

We acknowledge that the study is limited by its small sample size and that these results are preliminary, based on findings at two academic research institutions. However, the cohort evaluated in this study (n = 30) is largely similar to the individual arms in the OpACIN-NEO study and to other single-arm neoadjuvant immunotherapy trials[17,21,23–26]. With a median follow-up of 24 months, we also acknowledge that additional follow-up is needed to fully assess clinical impact and the durability of responses. However, this initial data is encouraging, and the pooled analyses of melanoma neoadjuvant trials support the importance of pathologic response rates as an early predictor of durable benefit[9]. Similarly, additional translational studies beyond the scope of this manuscript are planned, including RNA sequencing for broad assessment of additional immune signatures and populations that have been implicated in immunotherapy resistance[28,31].

In summary, neoadjuvant relatlimab and nivolumab is a highly active regimen that achieves a 70% pathologic response rate with a favourable safety profile in patients with high-risk, resectable clinical stage III or oligometastatic stage IV melanoma. These data are complementary to the RELATIVITY-047 study in patients with unresectable metastatic melanoma, and together further support the promise of this new combination immunotherapy regimen in this disease.

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

# Methods

## Patients

Eligible patients were 18 years or older with clinical stage III or oligometastatic (less than three organ sites with metastases) stage IV melanoma with lesions that were measurable by RECIST 1.1 (ref. [32]). Resectable clinical stage III melanoma was defined as clinically detectable, RECIST-measurable lymph node disease with or without regional in-transit or satellite metastases and without distant metastases. Resectability of stage III and IV disease was verified via multidisciplinary conference. Patients with recurrent melanoma or de novo American Joint Committee on Cancer 8th edition[3] clinical stage III or IV disease were considered eligible, and all melanoma subtypes, including uveal, mucosal or acral, were eligible for enrolment. All patients had Eastern Cooperative Oncology Group performance status of 0 or 1 with normal organ function and no contra-indication to surgery. Patients requiring active immunosuppressive therapy, or who had active autoimmune or infectious disease, or with uncontrolled cardiovascular disease or ongoing concurrent malignancy were excluded.

## Study design

This investigator-initiated, prospective study was conducted at two academic medical centres in the United States. Patients received two intravenous fixed doses of relatlimab 160 mg with nivolumab 480 mg at 4-week intervals. Surgery was planned 9 weeks after treatment initiation. Patients were given up to ten doses of the combination starting 4–6 weeks after surgery to complete a total of 12 doses. Patients were followed for 2 years postsurgery for any evidence of disease recurrence (study design details are provided in Fig. 1).

The primary end point was determination of pCR (defined as no viable tumour upon pathologic evaluation at surgery) rate[2]. For this exploratory biomarker study, a pathologic response rate of 30% was suggested for patients treated with this combination. Assuming this true pCR rate, the probability of at least 5 out of 30 patients experiencing a pCR is 0.97. Secondary end points included RECIST 1.1 overall response rate, safety, RFS, event-free survival, overall survival and correlation of immune profiling with response.

All patients were monitored for AEs according to the National Cancer Institute Common Terminology Criteria for Adverse Events, v.4.03 (ref. [33]). Due to concern for myocarditis based on prior relatlimab studies[1,20], patients were required to have cardiac troponin testing, in addition to assessment of blood counts, electrolytes, liver and kidney function before each scheduled infusion. All patients underwent baseline tumour staging (either computed tomography or positron-emission tomography-computed tomography of body and magnetic resonance imaging of brain) within 28 days of treatment initiation and again during week 8 for determination of RECIST response. Scans were performed every 3 months in the postoperative setting for up to 2 years after surgery. Core needle biopsy was performed within 28 days of treatment initiation and at weeks 3 and 5 for correlative research. Blood was collected at time of treatment initiation, weeks 3, 5, 9 and then every 12 weeks in the postoperative setting for up to 2 years (Fig. 1). Surgical resection was completed at week 9 per institutional standards and per the guidelines of the INMC[8,10]. Pathologic review of surgical resection specimens was performed by a small group of dermatopathologists who assessed the specimens according to the practices outlined by the INMC[2]. pCR was defined as no viable tumour, near pCR as greater than 0% but less than or equal to 10% viable tumour, pPR as greater than 10% to less than or equal to 50% viable tumour and pNR as greater than 50% viable tumour.

## Study oversight

The study was conducted in accordance with the clinical trial protocol and Good Clinical Practices Guidelines as defined by the International Conference on Harmonization and the Declaration of Helsinki. The study was approved by the institutional review boards of MD Anderson Cancer Center and Memorial Sloan Kettering Cancer Center. All patients provided informed consent for participation in the clinical trial. The study was designed by investigators at MD Anderson Cancer Center and the manuscript was written by the authors in its entirety. Trial monitoring was by the Investigational New Drugs office at MD Anderson Cancer Center. Study drugs were supplied by Bristol-Myers Squibb.

## Statistical analyses

RFS time was computed from surgery date to date of progression/recurrence or death (if died without progression/recurrence). Event-free survival time was computed from start of treatment to date of progression/recurrence or death (if died without progression/recurrence). Patients alive at the last follow-up date who did not experience progression/recurrence were censored. Patients who died without experiencing progression/recurrence were censored. Overall survival time was computed from start of neoadjuvant therapy to last known vital status. Patients alive at the last follow-up date were censored. The Kaplan–Meier method was used to estimate the outcome measures, and group differences were evaluated using the log-rank test. All statistical analyses were performed using SAS v.9.4 for Windows.

## Correlative studies

Blood and tumour were collected at the timepoints shown in Fig. 1. Cells were isolated and prepared from peripheral blood and tumour tissues for flow cytometry and CyTOF analyses as per the specifications below.

**Isolation and preparation of cells from peripheral blood and tissues.** Whole blood was collected in tubes containing sodium heparin (BD Vacutainer), resuspended in phosphate-buffered saline (PBS), layered atop Ficoll (StemCell Technologies) and centrifuged at 800$g$ for 25 min. The interface peripheral blood mononuclear cells (PBMCs) were harvested and washed twice with PBS and centrifuged at 500$g$ for 10 min. Fresh tumour tissue was dissociated with GentleMACS system (Miltenyi Biotec). PBMC and tumour specimens destined for CyTOF analysis were stained for viability with 5 µmol l$^{-1}$ cisplatin (Fluidigm, now Standard Biotools) in PBS containing 1% bovine serum albumin (BSA) and then washed three times. All specimens were resuspended in AB serum with 10% (vol/vol) dimethyl sulfoxide for storage in liquid nitrogen until downstream assays were performed.

## Flow cytometry staining and analysis

Flow cytometry analysis was performed on PBMCs (see Extended Data Table 3 for antibodies used in flow cytometry). Single-cell suspensions were stained with 16 fluorescent primary antibodies and live/dead dye. Specimens were analysed using the BD LSRFortessa ×20 cytometer and BD FACSDiva acquisition software v.8.0.1 (BD Biosciences), and downstream analyses were performed manually using FlowJo software v.10.5.3 (BD). See Extended Data Fig. 5 for flow cytometry sequential gating/sorting strategies.

## Mass cytometry staining and analysis

CyTOF analyses were performed on tumour specimens as well as PBMCs (see Extended Data Table 4 for antibodies used in CyTOF analysis). Single-cell suspensions were assayed with 41 antibodies, plus Ir DNA-intercalator and cisplatin. Antibodies were either purchased preconjugated from Fluidigm or purchased purified and conjugated in-house using MaxPar X8 Polymer kits (Fluidigm, now Standard Biotools). Briefly, samples were thawed and stained with cell surface antibodies in PBS containing 5% goat serum and 1% BSA for 30 min at 4 °C. Samples were then washed in PBS containing 1% BSA, fixed and permeabilized according to the instructions of the manufacturers using the FoxP3 staining buffer set (eBioscience), before being incubated with intracellular antibodies in permeabilization buffer for 30 min at 4 °C. Samples were washed and incubated in Ir intercalator (Fluidigm, now Standard Biotools) and stored at 4 °C until acquisition, generally

within 12 h. Immediately before acquisition, samples were washed and resuspended in water containing EQ 4 element beads (Fluidigm, now Standard Biotools). Samples were acquired on a Helios mass cytometer (Fluidigm, now Standard Biotools).

FCS files were preprocessed in R (R Foundation for Statistical Computing (https://www.R-project.org/)) using a CyTOF package (Premessa, Parker Institute for Cancer Immunotherapy (https://github.com/ParkerICI)) and gated manually in FlowJo (BD). Data were then exported as FCS files for downstream analysis and arcsinh transformed using a coefficient of 5 [$x\_transformed = arcsinh(x/5)$]. To visualize the high-dimensional data in two dimensions, the t-Distributed Stochastic Neighbor Embedding dimension reduction algorithm was applied, using all channels besides those used to manually gate the population of interest (for example, CD45 or CD3). Clustering analysis was performed in R using the FlowSOM and ConsensusClusterPlus packages[34].

## Graphics and statistics

Graphs were created and statistical analyses performed using GraphPad Prizm v.9.2 (GraphPad Software, LLC).

## Reporting summary

Further information on research design is available in the Nature Research Reporting Summary linked to this article.

## Data availability

Data supporting the findings of this study have been provided to *Nature* through direct deposition.

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

**Acknowledgements** Research reported in this publication was supported by the National Cancer Institute of the National Institutes of Health under grant award number P50CA221703. We would like to acknowledge the Melanoma Informatics, Tissue Resource and Translational Pathology Core at The University of Texas MD Anderson Cancer Center. This project was supported by the generous philanthropic contributions to The University of Texas MD Anderson Cancer Moon Shots Program. M. Postow is supported by Cancer Center Support Grant P30 CA08748 from the National Institutes of Health/National Cancer Institute. C.A. is supported by a pilot grant from the Parker Institute for Cancer Immunotherapy at MSKCC.

**Author contributions** R.N.A., E.M.B., M.A.D., J.A.W. and H.A.T. contributed to the conception and design of the study in collaboration with Bristol-Myers Squibb. R.N.A., M. Postow, M.I.R., I.C.G., L.S., J.L.M., M.K.W., J.E.G., J.E.L., R.P.G., E.Z.K., S.B.F., A.B.-W., A.N.S., M.C., D.C., E.K.B., D.B., P.M., S.P.P., A.D., A.L., M.A.D., J.A.W., C.A. and H.A.T. recruited and/or treated patients and gathered clinical data on efficacy and safety. D.R.M. analysed the clinical data and performed statistical analyses. F.D., C.N., A.G., X.Z., J.P.A. and P.S. performed biomarker analyses. M.T.T., C.T.-C., K.B. and V.G.P. performed pathologic analyses. B.R.K. and M. Patnana performed radiologic analyses. All authors interpreted the data. All authors had access to all the data in the study, participated in developing or reviewing the manuscript and provided final approval to submit the manuscript for publication.

**Competing interests** R.N.A.: research funding from Bristol-Myers Squibb, Iovance, Merck and Novartis; consulting role for Bristol-Myers Squibb, Iovance and Novartis. M. Postow: consulting fees from Aduro, Array BioPharma, Bristol-Myers Squibb, Eisai, Incyte, Merck, NewLink Genetics, Novartis and Pfizer; honoraria from Bristol-Myers Squibb and Merck; institutional support from Array BioPharma, AstraZeneca, Bristol-Myers Squibb, Infinity, Merck, Novartis and RGenix. M.I.R.: clinical research funding from Amgen; consulting/advisory board member role for Amgen, Castle BioSciences, Merck and Novartis. I.C.G.: research funding from Bristol-Myers Squibb, Merck and Pfizer; consulting role for Bristol-Myers Squibb and Novartis. J.L.M.: honoraria for Bristol-Myers Squibb and Roche; consultant for Merck. M.K.W.: advisory boards for Adagene, Bristol-Myers Squibb, Castle Biosciences, EMD-Serono, ExiCure, Merck, Pfizer and Regeneron. J.E.G.: consultant and/or advisory role; Merck and Regeneron. A.N.S.: research funding from Bristol-Myers Squibb, Checkmate Pharmaceuticals, Foghorn Therapeutics, Immunocore, Novartis, Pfizer, Polaris, Targovax and Xcovery; advisory board for Bristol-Myers Squibb, Immunocore and Novartis. A.D.: research funding from Apexigen, Idera and Nektar; consulting for Apexigen, Idera, Memgen, Nektar and Pfizer. S.P.P.: research funding from Bristol-Myers Squibb, Ideaya and Provectus; consulting honoraria from Cardinal Health, Castle Biosciences and Merck. M.A.D.: consultant to ABM Therapeutics, Apexigen, Array, Bristol-Myers Squibb, Eisai, GlaxoSmithKline, Pfizer, Roche/Genentech, Novartis, Sanofi-Aventis and Vaccinex; PI of research grants to GlaxoSmithKline, MD Anderson by Roche/Genentech, Merck, Myriad, Oncothyreon and Sanofi-Aventis. J.P.A.: consulting or stock ownership or advisory board for Achelois, Adaptive Biotechnologies, Apricity, BioAtla, BioNTech, Candel Therapeutics, Codiak, Dragonfly, Earli, Enable Medicine, Hummingbird, ImaginAb, Jounce, Lava Therapeutics, Lytix, Marker, PBM Capital, Phenomic AI, Polaris Pharma, Time Bioventures, Trained Therapeutix and Venn Biosciences. P.S.: consulting or stock ownership or advisory board for Achelois, Adaptive Biotechnologies, Affini-T, Apricity, BioAtla, BioNTech, Candel Therapeutics, Catalio, Codiak, Constellation, Dragonfly, Earli, Enable Medicine, Glympse, Hummingbird, ImaginAb, Infinity Pharma, Jounce, JSL Health, Lava Therapeutics, Lytix, Marker, MedImmune, Oncolytics, PBM Capital, Phenomic AI, Polaris Pharma, Sporos, Time Bioventures, Trained Therapeutix and Venn Biosciences. J.A.W.: compensation for speaker's bureau and honoraria from Bristol-Myers Squibb, Dava Oncology, Gilead, Illumina, Imedex, MedImmune, Omniprex, PeerView and Physician Education Resource; consultant/advisory board member for AstraZeneca, Biothera Pharmaceuticals, Bristol-Myers Squibb, GlaxoSmithKline, Merck, Micronoma, Novartis and Roche/Genentech. C.A.: consulting fees from Iovance. H.A.T.: research funding from GlaxoSmithKline; research funding and consulting honoraria from Bristol-Myers Squibb, Genentech, Merck and Novartis; consulting for Boxer, Eisai, Iovance, Karyopharm and Pfizer. All other authors report no conflicts of interest.

**Additional information**
**Correspondence and requests for materials** should be addressed to Rodabe N. Amaria.

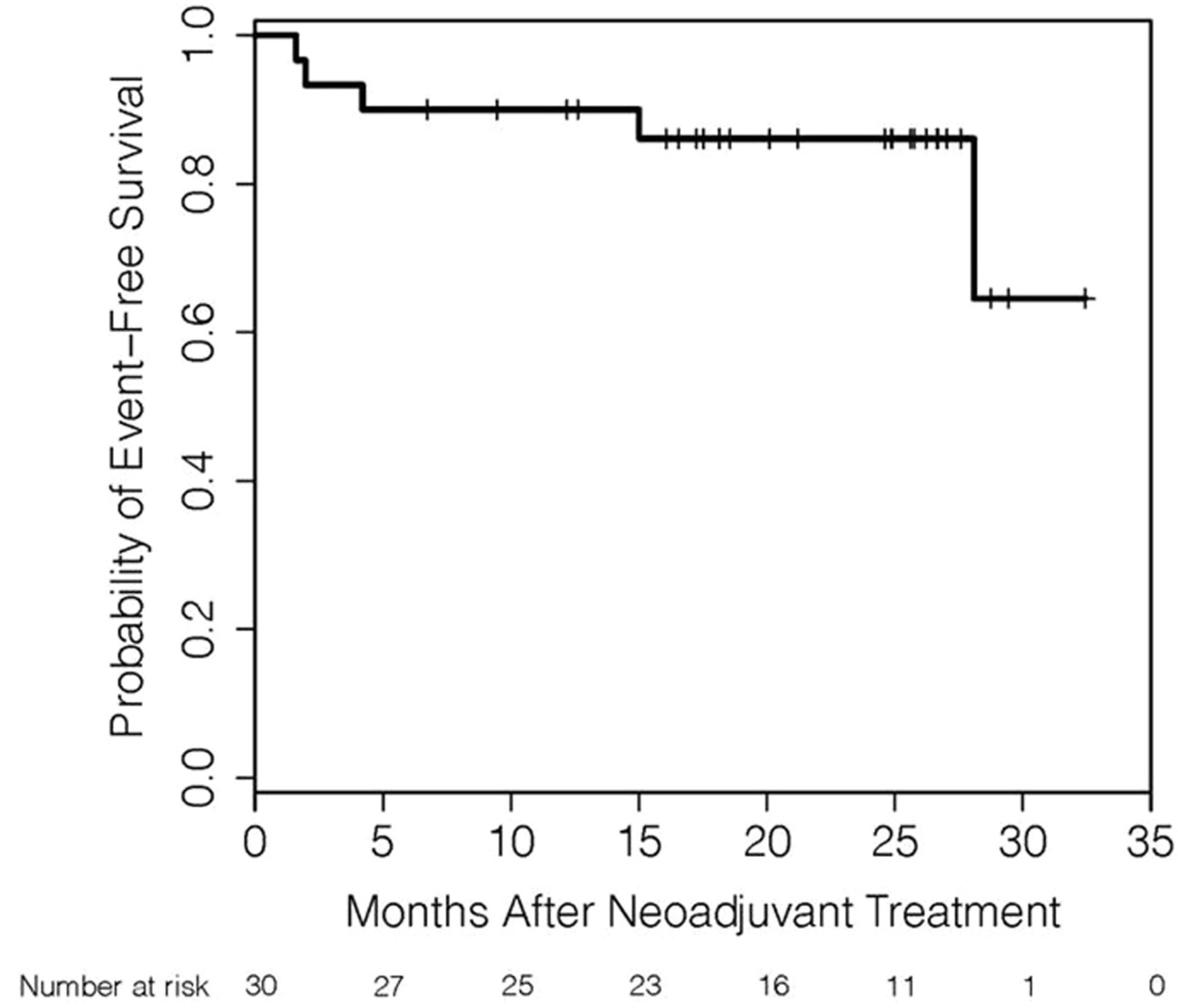

**Extended Data Fig. 1 | Probability of being event-free for all patients who received study treatment.**

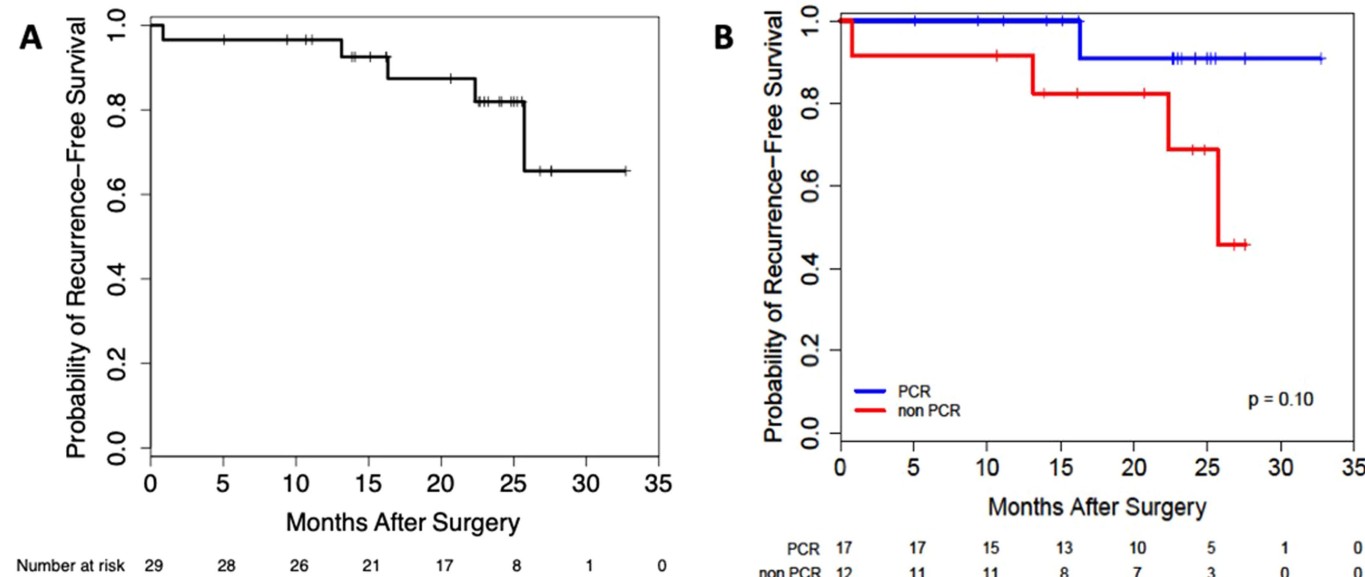

**Extended Data Fig. 2 | Probability of being recurrence-free. A**) Probability of being recurrence-free for all patients who underwent surgery. **B**) Probability of being recurrence-free based on pathologic complete response versus non-pathologic complete response (*P* = 0.10).

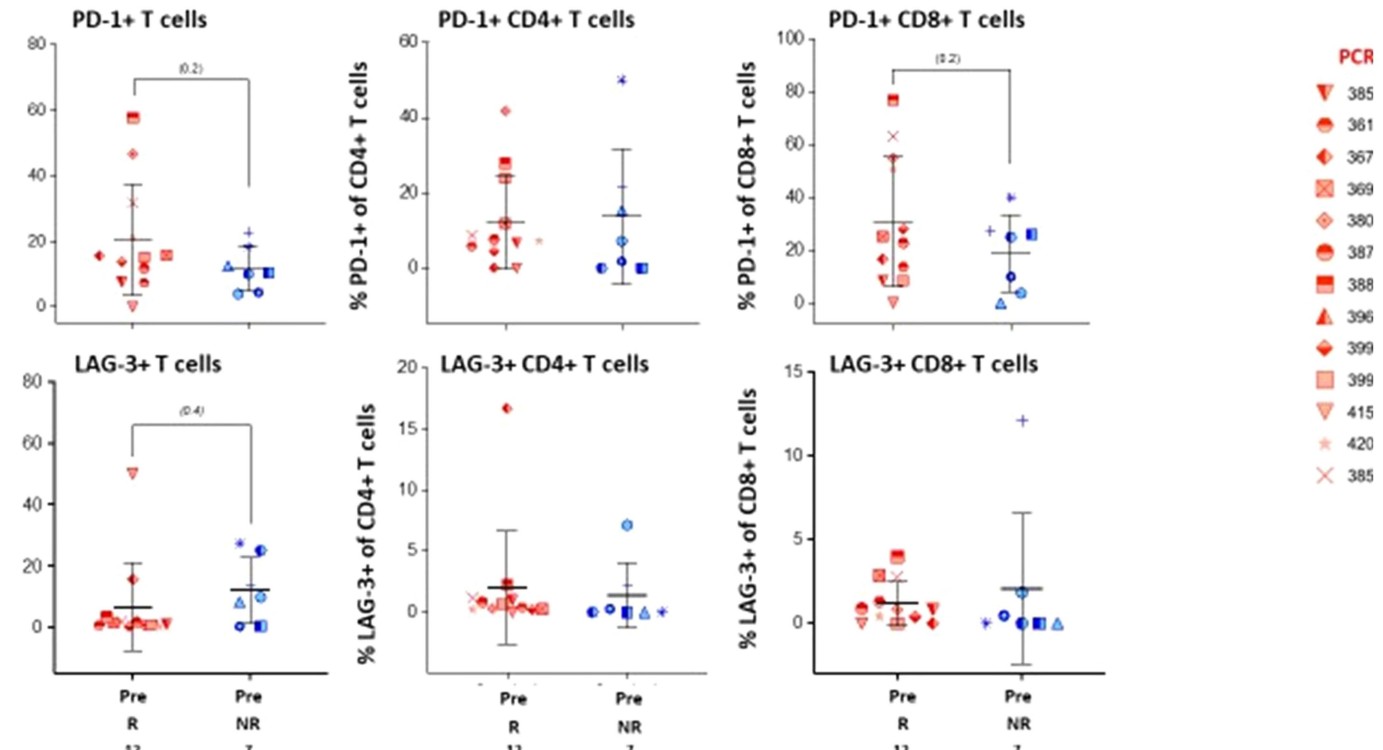

**Extended Data Fig. 3 | PD-1 and LAG-3 levels in baseline tumour.** Tumour infiltrating immune cells were assayed via CyTOF and analysed by manual gating for frequency of **A**) PD-1 and **B**) LAG-3 levels in T cells prior to treatment. Red, pathologic responders; blue, pathologic non-responders. Data are mean +/− SD; *P* values where shown were determined by two-tailed unpaired *t*-test, with no multiple comparisons. *n* values for each group/timepoint are indicated in each graph.

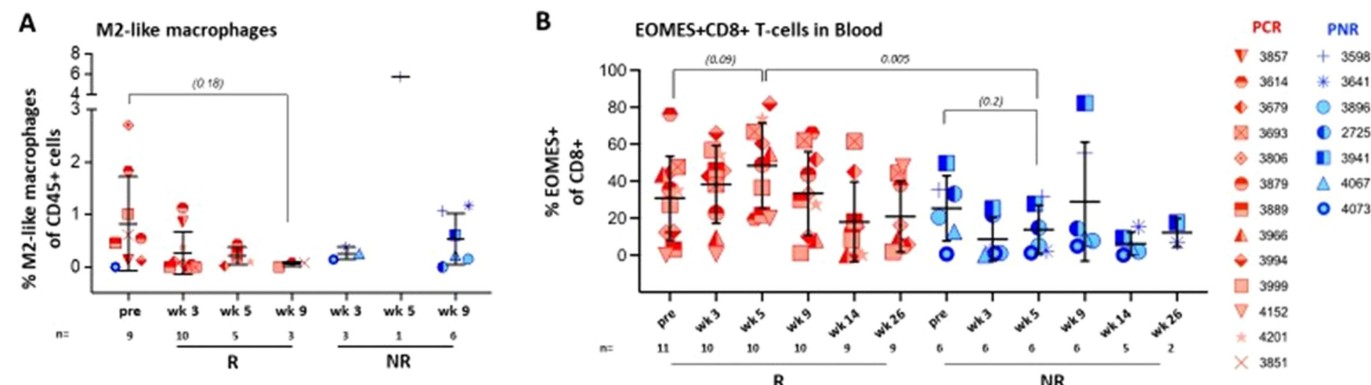

**Extended Data Fig. 4 | M2-like macrophages in tumour and EOMES+ CD8+ T cells in blood. A**) Frequency of an M2-like macrophage subset (CD68+ HLA-DR+ CD14+ VISTA+ CD163+ CD45RO+ PD-L1+) was determined by unsupervised clustering of CyTOF data from a single experiment. **B**) Frequency of EOMES+ CD8+ T cells. PBMCs isolated from blood samples were analysed by flow cytometry from a single experiment. Data are mean +/− SD; *P* value was determined by two-tailed unpaired *t*-test, with no multiple comparisons. *n* values for each group/timepoint are indicated in each graph. Red indicates pathologic responders; blue, non-responders.

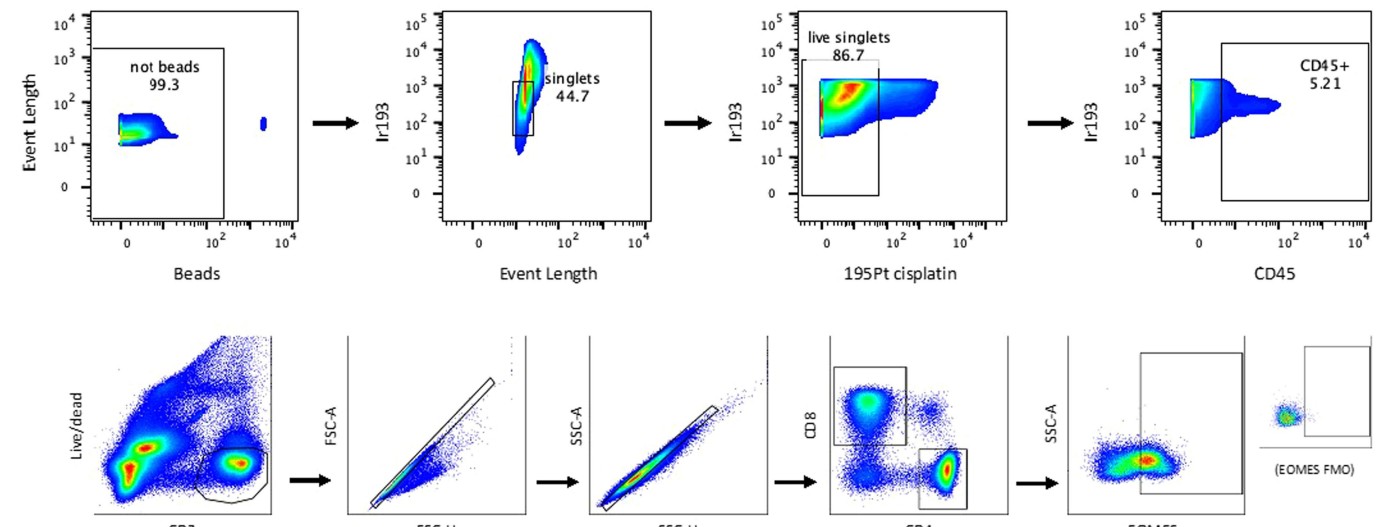

**Extended Data Fig. 5 | Gating schema for manual analysis of CyTOF data from tumour and blood specimens. A**) Tumours were mechanically dissociated and cells were stained with immune cell-specific antibodies. Specimens were assayed on the Helios mass cytometer via CyTOF Software. Cytometer data were then prepared for manual and unsupervised analyses via FlowJo software. Major cell populations were identified manually and reported. An example of one patient specimen is shown above for reference. **B**) Gating schema for flow cytometric analysis of blood specimens. Peripheral blood mononuclear cells from patient specimens were stained along with FMO (fluorescence minus one) controls and assayed via a BD LSRFortessa cytometer and BD FACSDiva acquisition software. Data were analysed via FlowJo software as described above. Briefly, live CD3+ singlets were identified and gated into T cell lineages, and those lineages analysed for frequency of each of eight phenotypic markers (BCL6, BLIMP1, CD27, CD28, cMYC, EOMES, ICOS, Ki67) as defined by FMO (fluorescence minus one) specimens. An example of one phenotypic marker (EOMES) in one patient specimen is shown above for reference.

**Extended Data Table 1 | Baseline patient characteristics**

| | Total cohort (n=30) |
|---|---|
| Age | |
| Median (range) | 60 (35-79) |
| Gender | |
| Female / Male | 11 (37%) / 19 (63%) |
| ECOG PS | |
| 0 / 1 | 28 (93%)/ 2 (7%) |
| Clinical stage* | |
| IIIB | 18 (60%) |
| IIIC | 8 (26%) |
| IIID | 2 (7%) |
| IV M1a | 2 (7%) |
| LDH above upper limit of normal | 3 (10%) |
| BRAF V600E/K mutated | 5 (17%) |
| Pretreatment status | |
| Prior surgery | 20 (67%) |
| Prior systemic therapy | 1 (3%) |
| Median target lesions sum of diameters | 26 (13-76 mm) |

**Extended Data Table 2 | Immune-related adverse events during neoadjuvant and adjuvant therapy**

| Immune Related Adverse Events | Neoadjuvant Treatment (n=30) | | Adjuvant Treatment (n=27) | |
|---|---|---|---|---|
| | Grade 1-2 (%) | Grade 3-4 (%) | Grade 1-2 (%) | Grade 3-4 (%) |
| Adrenal insufficiency | 1 (3%) | 0 | 3 (11%) | 3 (11%) |
| Increased ALT/AST | 3 (10%) | 0 | 8 (30%) | 2 (7%) |
| Increased alkaline phosphatase | 0 | 0 | 2 (7%) | 1 (4%) |
| Anemia | 3 (10%) | 0 | 8 (30%) | 0 |
| Anorexia | 1 (3%) | 0 | 4 (15%) | 0 |
| Arthralgia | 0 | 0 | 3 (11%) | 2 (7%) |
| Troponin increase | 3 (10%) | 0 | 3 (11%) | 0 |
| CPK increase | 2 (7%) | 0 | 1 (4%) | 1 (4%) |
| Creatinine increase | 2 (7%) | 0 | 4 (15%) | 0 |
| Diarrhea | 0 | 0 | 4 (15%) | 0 |
| Hypothyroidism | 2 (7%) | 0 | 6 (22%) | 0 |
| Fatigue | 5 (17%) | 0 | 7 (26%) | 0 |
| Hyponatremia | 3 (10%) | 0 | 5 (19%) | 2 (7%) |
| Infusion reaction | 2 (7%) | 0 | 0 | 0 |
| Myalgia | 0 | 0 | 4 (15%) | 0 |
| Nausea | 1 (3%) | 0 | 4 (15%) | 0 |
| Rash | 5 (17%) | 0 | 5 (19%) | 0 |

**Extended Data Table 3 | Antibodies for flow cytometry analysis**

| Antibody | Clone | Vendor | Catalog # |
|---|---|---|---|
| CD45RO | UCHL1 | BioLegend | 304218 |
| CD28 | CD28.2 | eBioscience | 47-0289-42 |
| CD62L/L-selectin | DREG-56 | BioLegend | 304828 |
| Yellow Live/Dead | n/a | Life Technologies | L34959 |
| CD45RA/PTPRC | HI100 | BioLegend | 304136 |
| CD197/CCR7 | G043H7 | BioLegend | 353230 |
| CD8a | RPA-T8 | eBioscience | 58-0088-42 |
| CD3 | UCHT1 | BD Biosciences | 562280 |
| CD27 | O323 | eBioscience | 15-0279-42 |
| CD4 | SK3 | eBioscience | 35-0047-42 |
| CD278/ICOS | ISA-3 | eBioscience | 25-9948-42 |
| Eomes | WD1928 | eBioscience | 50-4877-42 |
| BLIMP-1 | 646702 | R&D Systems | IC36081G |
| BCL-6 | K112-91 | BD Biosciences | 562198 |
| T-bet | 4B10 | BioLegend | 644817 |
| Ki-67 | Ki-67 | BioLegend | 350516 |
| cMyc | 9E10 | R&D Systems | IC3696P |

**Extended Data Table 4 | Antibodies for CyTOF analysis**

| Antibody | Clone | Vendor | Catalog Number |
|---|---|---|---|
| CD45 | HI30 | Fluidigm | 3089003B |
| CD19 | HIB19 | BioLegend | 302247 |
| CD4 | RPA-T4 | BioLegend | 300541 |
| CD8 | RPA-T8 | BioLegend | 301053 |
| CD163 | GHI/61 | BioLegend | 333602 |
| CD14 | M5E2 | BioLegend | 301843 |
| CCR7 | G043H7 | BioLegend | 353237 |
| PD-1 | EH12.2H7 | BioLegend | 329941 |
| Eomes | WD1928 | eBioscience | 14-4877-82 |
| CD1c | L161 | BioLegend | 331502 |
| CD11c | 3.9 | Fluidigm | 3146014B |
| T-bet | 4B10 | BioLegend | 644825 |
| CD16 | 3G8 | Fluidigm | 3148004B |
| LAG-3 | 874501 | R&D | MAB23193 |
| PD-L1 | MIH1 | eBioscience | 14-5983-82 |
| CD123 | 6H6 | Fluidigm | 3151001B |
| TCRγδ | 11F2 | Fluidigm | 3152008B |
| ICOS | ISA-3 | eBioscience | 14-9948-82 |
| TIGIT | MBSA43 | Fluidigm | 3154016B |
| CD45RA | H100 | Fluidigm | 3155011B |
| CD86 | IT2.2 | Fluidigm | 3156008B |
| LAG-3 | 11C3C65 | BioLegend | 369302 |
| CD161 | HP3G10 | Fluidigm | 3159004B |
| CD141 | AD5-14H12 | Miltenyi Biotec | 130-090-694 |
| CTLA-4 | 14D3 | Fluidigm | 3161004B |
| FOXP3 | PCH101 | Fluidigm | 3162011A |
| CRTH2 | BM16 | Fluidigm | 3163003B |
| CXCR5 | RF8B2 | Fluidigm | 3164029B |
| TCF7 | 7F11A10 | BioLegend | 655202 |
| TIM3 | F38-2E2 | BioLegend | 345019 |
| BTLA | J168-540 | BD | 624084 |
| CD73 | AD2 | Fluidigm | 3168015B |
| CCR10 | 314305R | R&D Systems | MAB3478R-100 |
| CD3 | UCHT1 | BioLegend | 300437 |
| CD68 | Y1/82A | Fluidigm | 3171011B |
| CD28 | CD28.2 | BioLegend | 302937 |
| Granzyme B | GB11 | Fluidigm | 3173006B |
| Ki67 | Ki67 | BioLegend | 350523 |
| CD45RO | UCHL1 | BioLegend | 304239 |
| CD56 | NCAM16.2 | Fluidigm | 3176008B |
| HLA-DR | L243 | Fluidigm | custom |

# Reporting Summary

## Statistics

For all statistical analyses, confirm that the following items are present in the figure legend, table legend, main text, or Methods section.

| n/a | Confirmed | |
|---|---|---|
| ☐ | ☒ | The exact sample size (*n*) for each experimental group/condition, given as a discrete number and unit of measurement |
| ☐ | ☒ | A statement on whether measurements were taken from distinct samples or whether the same sample was measured repeatedly |
| ☐ | ☒ | The statistical test(s) used AND whether they are one- or two-sided *Only common tests should be described solely by name; describe more complex techniques in the Methods section.* |
| ☐ | ☒ | A description of all covariates tested |
| ☐ | ☒ | A description of any assumptions or corrections, such as tests of normality and adjustment for multiple comparisons |
| ☐ | ☒ | A full description of the statistical parameters including central tendency (e.g. means) or other basic estimates (e.g. regression coefficient) AND variation (e.g. standard deviation) or associated estimates of uncertainty (e.g. confidence intervals) |
| ☐ | ☒ | For null hypothesis testing, the test statistic (e.g. *F*, *t*, *r*) with confidence intervals, effect sizes, degrees of freedom and *P* value noted *Give P values as exact values whenever suitable.* |
| ☒ | ☐ | For Bayesian analysis, information on the choice of priors and Markov chain Monte Carlo settings |
| ☒ | ☐ | For hierarchical and complex designs, identification of the appropriate level for tests and full reporting of outcomes |
| ☒ | ☐ | Estimates of effect sizes (e.g. Cohen's *d*, Pearson's *r*), indicating how they were calculated |

*Our web collection on statistics for biologists contains articles on many of the points above.*

## Software and code

Policy information about availability of computer code

| Data collection | Clinical data collection via MDACC Prometheus Data Collection System. Flow cytometry data collection software: BD FACSDiva software, version 8.0.1 for flow cytometry (Becton Dickinson & Company) CyTOF data collection software: CyTOF Software v 7.0.8493 for mass cytometry (Fluidigm, now Standard Biotools) |
|---|---|
| Data analysis | Clinical data analysis by SAS 9.4 by Windows  (Copyright © 2002-2012 by SAS Institute Inc., Cary, NC) FlowJo v.10.5.3 (Becton Dickinson & Company); WorkFlow script for unsupervised clustering (Nowicka et al F1000 Research 2017), running on R version 3.5.2 (R Foundation for Statistical Computing) R Foundation for Statistical Computing |

For manuscripts utilizing custom algorithms or software that are central to the research but not yet described in published literature, software must be made available to editors and reviewers. We strongly encourage code deposition in a community repository (e.g. GitHub). See the Nature Portfolio guidelines for submitting code & software for further information.

## Data

Policy information about availability of data

All manuscripts must include a data availability statement. This statement should provide the following information, where applicable:
- Accession codes, unique identifiers, or web links for publicly available datasets
- A description of any restrictions on data availability
- For clinical datasets or third party data, please ensure that the statement adheres to our policy

Full data to be provided upon request, there are no restrictions

# Field-specific reporting

Please select the one below that is the best fit for your research. If you are not sure, read the appropriate sections before making your selection.

☒ Life sciences ☐ Behavioural & social sciences ☐ Ecological, evolutionary & environmental sciences

For a reference copy of the document with all sections, see nature.com/documents/nr-reporting-summary-flat.pdf

# Life sciences study design

All studies must disclose on these points even when the disclosure is negative.

| | |
|---|---|
| Sample size | Sample size justification provided in the methods of the manuscript. |
| Data exclusions | There were no data exclusions |
| Replication | Data was not able to be replicated as this is a study using human subjects and data generated is unique to the study subject |
| Randomization | This was a single arm study with no randomization |
| Blinding | Blinding was not utilized as this is a single arm, non-randomized trial |

# Reporting for specific materials, systems and methods

We require information from authors about some types of materials, experimental systems and methods used in many studies. Here, indicate whether each material, system or method listed is relevant to your study. If you are not sure if a list item applies to your research, read the appropriate section before selecting a response.

### Materials & experimental systems

| n/a | Involved in the study |
|---|---|
| ☐ | ☒ Antibodies |
| ☒ | ☐ Eukaryotic cell lines |
| ☒ | ☐ Palaeontology and archaeology |
| ☒ | ☐ Animals and other organisms |
| ☐ | ☒ Human research participants |
| ☐ | ☒ Clinical data |
| ☒ | ☐ Dual use research of concern |

### Methods

| n/a | Involved in the study |
|---|---|
| ☒ | ☐ ChIP-seq |
| ☐ | ☒ Flow cytometry |
| ☒ | ☐ MRI-based neuroimaging |

## Antibodies

| | |
|---|---|
| Antibodies used | The anti PD-1 antibody nivolumab and anti LAG-3 antibody relatlimab were provided by the study sponsor Bristol-Myers Squibb as part of their investigational supply of agents. Relatlimab should be stored at 2°C to 8°C (36oF to 46oF) with protection from light. Do not freeze the drug product.<br>Relatlimab is to be administered combined with nivolumab in the same bag as a 60 minute IV infusion through a 0.2/1.2-☐m pore size, low-protein-binding polyethersulfone membrane in-line filter at the protocol-specified doses. The Relatlimab and nivolumab injection can be diluted with 0.9% sodium chloride injection (normal saline), to protein concentrations no lower than 1.33 mg/mL. Detailed instructions for drug product dilution and administration are provided in the pharmacy manual for the clinical study |
| Validation | These antibodies were provided as part of BMS's investigational study supply |

## Human research participants

Policy information about studies involving human research participants

| | |
|---|---|
| Population characteristics | Clinical stage III melanoma with resectable disease. Patients aged 18 and over, ECOG PS 0-1, normal organ function with no contra-indications to surgery. |
| Recruitment | Patients were enrolled at MD Anderson and Memorial Sloann Kettering in the Melanoma Clinics. Patients were offered either clinical trial enrollment or standard of care therapies. Patients were provided copies of the study informed consent document and were fully aware of risks prior to trial enrollment. As patients needed to fulfill inclusion criteria of trial, this could have caused selection bias. |
| Ethics oversight | IRB of MDACC and MSKCC provided ethics oversight. An informed consent statement was included in the Methods/Study Oversight section |

Note that full information on the approval of the study protocol must also be provided in the manuscript.

# Clinical data

Policy information about clinical studies

All manuscripts should comply with the ICMJE guidelines for publication of clinical research and a completed CONSORT checklist must be included with all submissions.

| | |
|---|---|
| Clinical trial registration | NCT02519322 |
| Study protocol | Available upon request |
| Data collection | 9/19/2018 - 9/23/2020 for enrollment, patients followed for at least 1 year after date of last enrollment. Data was stored in a secure database sponsored by MD Anderson Cancer Center and was able to be accessed by staff at MSKCC for direct data input. |
| Outcomes | The primary outcome was assessment of pathologic response following neoadjuvant therapy as per the criteria of the INMC which is agreed upon pathologic response criteria utilized in melanoma neoadjuvant studies. Secondary outcomes including RECIST response, safety, RFS, EFS, and OS are standard outcome criteria utilized in neoadjuvant studies to describe characteristics of response. Correlation of immune profiling with response was exploratory in nature and dependent upon results of correlative studies. |

# Flow Cytometry

## Plots

Confirm that:

☒ The axis labels state the marker and fluorochrome used (e.g. CD4-FITC).

☒ The axis scales are clearly visible. Include numbers along axes only for bottom left plot of group (a 'group' is an analysis of identical markers).

☒ All plots are contour plots with outliers or pseudocolor plots.

☒ A numerical value for number of cells or percentage (with statistics) is provided.

## Methodology

| | |
|---|---|
| Sample preparation | Sample preparation details can be found in the Methods section of the manuscript. Isolation and preparation of cells from peripheral blood and tissues<br>Whole blood was collected in tubes containing sodium heparin (BD Vacutainer), resuspended in PBS, layered atop Ficoll (StemCell Technologies) and centrifuged at 800 × g for 25 minutes. The interface peripheral blood mononuclear cells (PBMC) were harvested and washed twice with PBS and centrifuged at 500 × g for 10 minutes. Fresh tumor tissue was dissociated with GentleMACS system (Miltenyi Biotec). PBMC and tumor specimens destined for CyTOF analysis were stained for viability with 5 µmol/L cisplatin (Fluidigm) in PBS containing 1% BSA and then washed 3x. All specimens were resuspended in AB serum with 10% (vol/vol) DMSO for storage in liquid nitrogen until downstream assays were performed. |
| Instrument | BD LSRFortessa x20 flow cytometer (Becton Dickinson & Company);<br>Helios mass cytometer (Fluidigm, now Standard Biotools) |
| Software | Acquisition Software: BD FACSDiva software, version 8.0.1 for flow cytometry (Becton Dickinson & Company); CyTOF Software v 7.0.8493 for mass cytometry (Fluidigm, now Standard Biotools)<br>Analysis Software: FlowJo v.10.5.3 (Becton Dickinson & Company); WorkFlow script for unsupervised clustering (Nowicka et al F1000 Research 2017), running on R version 3.5.2 (R Foundation for Statistical Computing) |
| Cell population abundance | Sorting was not performed. Bulk tumor cells were procured, the immune fraction enriched via buffy layer, and immune cell-specific detection antibodies were used for cytometry analysis. |
| Gating strategy | Please see Extended Data Figure 5 to review the Flow Cytometry and CyTOF gating strategies. |

☒ Tick this box to confirm that a figure exemplifying the gating strategy is provided in the Supplementary Information.

