## [Peer Review File · Nature]

Manuscript Title: Neoadjuvant Relatlimab and Nivolumab in Resectable Melanoma

Reviewer Comments & Author Rebuttals

Reviewer Reports on the Initial Version:

Referees' comments:

Referee #1 (Remarks to the Author):

A. This is a single-arm, neoadjuvant study at two centers examining combination PD-1/LAG-3 blockade. The trial demonstrates a high-response rate with acceptable toxicity, albeit with relatively short follow up at this point. The neoadjuvant paradigm is of interest in melanoma currently and this combination is of interest given the recent documented utility in unresectable metastatic disease.

B. The trial design follows current recommendations and examines a combination that is novel in the neoadjuvant setting.

C. The trial methodology appears sound. There are specific questions regarding the correlative statistics. The sample size and duration of follow up limit the significance of the observations.

D. Could the authors clarify the statistical approach to the correlative analyses? It appears most of these comparisons were not hypothesis-driven (i.e. pre-specified) or validated. Specifically, were any adjustments made for multiple comparisons? Were any of the observations noted in the manuscript (e.g. decrease in M2 macrophages) statistically significant, bearing those considerations in mind?

E. Conclusions: The authors note the hazards of making cross-trial comparisons, but then go on to make such comparisons. They conclude based on those comparisons that the currently reported combination is superior to single agent PD-1 blockade, and equal to flipped-dose ipi/nivo with less toxicity.

While it is tempting to make that type of assertion, these conclusions seem too strong given potentially significant differences in the populations treated in different studies (e.g. the extent of disease in the current trial seems very low (sum of diameters of 26 mm, 60% Stage IIIB, 90% normal LDH)).

F. Improvements: Clarification of the following would be helpful:

1) Were any surgical quality control assessments done in the resection specimens? Current INMC recommendations are for full therapeutic dissection unless a protocol specifies otherwise.

2) Two "local" recurrences are noted. Were these local to the metastatic site or the primary? These are noted to have occurred 3 and 14 months after completing adjuvant therapy. Could the authors clarify the temporal relationship to surgery (i.e. did those patients get 10 doses of adjuvant therapy?)

3) The Menzies meta-analysis suggests overall and melanoma-specific survival in responders to

neoadjuvant immunotherapy (there PD-1 monotherapy or ipi/nivo) are close to 100%. Although the numbers are small in the current study, 1 patient (8%) who had a pCR has died of metastatic melanoma, even with only the current early follow up. This raises the question of whether pathologic response (which does seem higher with this combination) is an unreliable surrogate endpoint to reflect OS or MSS benefit. Do the authors feel this argues for caution in relying too heavily on pCR to make decisions regarding regulatory approval or treatment preferences?

G. References: No concerns

H. Clarity: the manuscript is well written

Referee #2 (Remarks to the Author):

This article reports on a single arm clinical trial testing the ability of a combination of anti-PD-1 and anti-LAG3 antibody therapy to induce pathologic responses in patients with surgically resectable melanoma.

Main comments:

What was the statistical design to result in 30 subjects being accrued, what was the target pathological response and the power of this trial to test the study hypothesis?

Were there different melanoma subtypes eligible? In particular, were there patients with acral, mucosal or desmoplastic melanoma included in this series?

In the Abstract, the authors compare the relapse-free survival of patients with a biopsy with pathological response or no pathological response, and provide a p value. However, this p value is a result of non-randomized groups, and is prone to confounding effects of variables other than response to therapy.

The Introduction is unnecessarily too long to put this study in context and justify its conduct.

There were 30 patients who initiated neoadjuvant treatment, and this should be the denominator for all efficacy and safety assessments. However, it seems like the pathological response rates are based on 29 patients, omitting a patient who progressed on therapy.

The first paragraph of the Discussion is an unnecessary numerical repetition of the results that are just immediately before presented in the Results section. It would be desirable to shorten this paragraph to at most a couple of sentences describing the top line results without repeating all the numbers. As with the Introduction, the Discussion seems too long for the data being presented.

Minor comments:

One subject who consented to participate was not enrolled due to “insurance denial”. However, the clinical trial provided the study drugs. What insurance denial led to inability to participate?

One subject is reported to having passed away 14 months after surgery with an unconfirmed disease progression in the brain (lines 208-209). Is this “unconfirmed” due to an unclear lesion, or because there were not two serial scans, or because there was no biopsy? If the patient develop brain metastases by radiologic assessment and died of progressive disease, the point of being “unconfirmed” is not relevant.

“NR patients” and “R patients” in lines 230-232 seem to refer to patients whose pathological assessment had a response or response to therapy. It would be desirable to not make the patients an adjective of their tumor response.

Referee #3 (Remarks to the Author):

Amaria et al. reported a study with relatlimab and nivolumab as neoadjuvant for patients with resectable stage III or oligometastatic stage IV melanoma. Patients received 2 neoadjuvant doses followed by surgery, and then 10 doses of adjuvant combination therapy. The use of checkpoint inhibitors combinations is an important strategy to overcome resistance and improve survival. In particular, the anti-LAG-3 relatlimab in combination with nivolumab showed positive results in a recent phase III study in metastatic melanoma patients (Tawbi et al NEJM 2022) with a subsequent approval from FDA. The study from Amaria et al. is the first which evaluated an anti-LAG-3 in combination with anti-PD-1 as neoadjuvant therapy. The data looks certainly interesting in comparison with all the data already published in this field. Thirty patients were enrolled and the combination resulted in 59% pCR rate and 73% overall pathological response rate. The safety was consistent with all the previous reports from this new combination (Ascierto et al ESMO 2017; Tawbi et al NEJM 2022) showed and incidence of AEs G3-4 slightly higher compared to anti-PD-1 monotherapy. Excellent results if we consider that with the combination ipilimumab/nivolumab the pCR rate was 50-60% in previous reports with a surely higher incidence of side effects.

However, if I certainly agree about the more favorable safety profile, in term of efficacy at the moment this data should be considered preliminary results since come from a small cohort of patients. Moreover, if the pCR is the direct effect of the neoadjuvant treatment, the RFS may be affected by the subsequent adjuvant treatment. Indeed, the subsequent adjuvant therapy with the combination could be a confounding factor to define the efficacy of the neoadjuvant strategy. This aspect should be discussed in the manuscript.

In the manuscript it should also be discussed the risk that results could be affected by patients selection. The fact that the majority of patients were BRAF wild type and the median size of the lesions was 26 mm might indicate a selection bias. If it's true that the Authors described an absence of correlation between the tumor lesions size and response, the small number of patients enrolled doesn't allow to make any conclusion. These are surely interesting data but still preliminary.

As regard to the correlative studies, flow cytometry and mass cytometry (CyTOF) analyzes were performed on tumor specimens as well as PBMCs. Biomarker analysis showed a high frequency of CD45+ in the pretreatment responders' samples compared to pretreatment samples of non-responder and an increase of effector CD8+ T cell subset and a memory CD4+ T cell subset was seen in responding versus non responding.

Another interesting evidence is the decrease of M2 macrophages in patients with pathologic response in TME over the course of neoadjuvant therapy. Unfortunately, it is not described clearly the methodology of this finding. M2 detection is usually evaluated with CD68 +, CD206+, CD163+ and for some studies also PDL1+. In this work, the identification of M2 occurs with several different receptors which could select a larger population of cells not specifically M2.

Authors should better discuss this aspect.

Finally, it would be interesting to get data from a complete mutational analysis by NGS on both pretreatment and posttreatment tumor tissue.

Referee #4 (Remarks to the Author):

This is a straightforward single arm trial, and another good attempt to find effective neoadjuvant therapies with modern drugs.

1. The design parameters are missing from the paper: what was the target pCR rate and what background rate was assumed in the sample size calculation? These are essential in order to see whether the target was achieved. Also, an appropriate confidence interval needs to be provided for the pCR rate.

2. There needs to be Kaplan-Meier curves for EFS, RFS and OS (in the appendix) for all patients (in addition to showing curves split by response). In the Discussion, say how the observed results for these outcomes compare with those seen in patients in routine practice?

3. The definitions of RFS and EFS were unclear. It is usual that death from any cause is counted as an event; is this correct in this study? The text states that "patients who died without experiencing progression/recurrence were censored", which implies that someone who died from melanoma (without progression) would be censored – but this should not be the case, they should be counted as an event (there are several patients like this in clinical trials, it just means they died before progression could be clinically ascertained). Also, for OS, patients who have not died should be censored at the date of last follow up (unclear what 'last known vital status' actually means).

4. Could the high pCR rate be partly influenced by all patients coming from 2 highly expert cancer centres, nearly all patients were ECOG 0, or 60% stage IIIb?

Author Rebuttals to Initial Comments:

Referee #1 (Remarks to the Author):

A. This is a single-arm, neoadjuvant study at two centers examining combination PD-1/LAG-3 blockade. The trial demonstrates a high-response rate with acceptable toxicity, albeit with relatively short follow up at this point. The neoadjuvant paradigm is of interest in melanoma currently and this combination is of interest given the recent documented utility in unresectable metastatic disease.

Thank you for your assessment of the trial data and positive assessment of the value of neoadjuvant therapy in melanoma.

B. The trial design follows current recommendations and examines a combination that is novel in the neoadjuvant setting.

Thank you for your comment. We designed this trial in accordance with the recommendations set forth from the International Neoadjuvant Melanoma Consortium. We agree the combination of relatlimab and nivolumab is novel and is the first trial to use these agents in the neoadjuvant setting.

C. The trial methodology appears sound. There are specific questions regarding the correlative statistics. The sample size and duration of follow up limit the significance of the observations.

Thank you for this feedback. Your specific questions on the correlative statistics are addressed in the point below. We agree the sample size of 30 patients does limit the statistical power of the observations, however the data herein remain significant and are complementary to data generated by the much larger RELATIVITY-047 study. We also agree the follow-up time in this trial is a limitation, however both the sample size and follow-up time are comparable to other published neoadjuvant therapy trial data in melanoma.

D. Could the authors clarify the statistical approach to the correlative analyses? It appears most of these comparisons were not hypothesis-driven (i.e. pre-specified) or validated. Specifically, were any adjustments made for multiple comparisons? Were any of the observations noted in the manuscript (e.g. decrease in M2 macrophages) statistically significant, bearing those considerations in mind?

Thank you for these valuable questions. For the correlative analyses, we have looked at frequencies of total tumor infiltrating leukocytes (CD45+ TILs), major immune cell subsets (CD4+, CD8+ T-cells etc.), as well as other specific immune cell subsets defined by our Flow Cytometry and CyTOF panels. Most comparisons we included in the manuscript were hypothesis-driven. In Figure 4, the frequency of CD45, Effector CD8 cells, and Memory CD4+ cells was reported. We used unpaired t-tests to evaluate changes in frequencies between R and NR at baseline, as well as baseline and treated samples, in both responders and non-responders. Our null hypotheses state that responders will not have higher frequency of CD45+ at baseline and will not have increased frequencies in effector CD8 and memory CD4 with treatment. We used p values < 0.05 in order to reject those null hypotheses and concluded that frequency of CD45 is higher in responders at baseline, and responders experience a significant increase of tumor-associated effector CD8 and memory CD4 cells after treatment, while non-responders do not. In Supplementary Figure 2, we used unpaired t-tests to compare frequency of tumor-

associated PD-1+ and LAG-3+ T lymphocytes between responders and non-responders at baseline. We hypothesized that responders might have more LAG3+ cells at baseline. P values > 0.05 indicate no difference in frequencies of these populations between responders and non-responders.

In Supplementary Figure 3A, we performed unsupervised data analysis on CyTOF data, and report on an M2-like macrophage population in tumor specimens. This approach was hypothesis-generating. We used an unsupervised t-test in order to compare each treated specimen independently to the baseline specimen, within responders and non-responders. We can only report that, with a p value = 0.18, there is only a trend in which these cells decrease with treatment in responders but do not change in non-responders. Many other cell clusters were identified using this unsupervised analysis technique but were not reported, as we found them to be neither statistically nor biologically significant. In Supplementary Figure 3B, we then pursued a novel observation taken from the CyTOF data to report on an EOMES+ CD8+ cell population in blood specimens. We went back and performed manual analysis of flow cytometry data from blood to look for this population. We used unpaired t-tests to compare baseline specimens to each individual post-treatment specimen (first within responders, and then within non-responders), and we also used unpaired t-tests to compare between responders and non-responders at each individual timepoint. P values of 0.09 in responders and 0.2 in non-responders suggest only a trend toward an increase in responders and a decrease in non-responders over time. However, with a p value of 0.005, we do see a significant difference in EOMES+ cells between responders and non-responders, at week 5.

In all figures, we did not make adjustments for multiple comparisons, and p values for all figures were calculated using unpaired t-tests.

E. Conclusions: The authors note the hazards of making cross-trial comparisons, but then go on to make such comparisons. They conclude based on those comparisons that the currently reported combination is superior to single agent PD-1 blockade, and equal to flipped-dose ipi/nivo with less toxicity.

While it is tempting to make that type of assertion, these conclusions seem too strong given potentially significant differences in the populations treated in different studies (e.g. the extent of disease in the current trial seems very low (sum of diameters of 26 mm, 60% Stage IIIB, 90% normal LDH)).

Thank you for this important point and perspective. We have re-written this portion of the manuscript in the Discussion Section to read:

The INMC has demonstrated clinical utility of neoadjuvant checkpoint blockade including the safety profile and impact on pCR of single agent anti-PD-1, as well as its combination with anti-CTLA4^{7, 10, 14-15, 21}. Single agent anti-PD1 resulted in pCR rates of 20-30% and IRAE rates of ~15%^{14,21}. The combination of ipilimumab and nivolumab induced pCR rates of nearly 50%, however, the high dose regimen (ipilimumab 3 mg/kg + nivolumab 1 mg/kg) is considered overly toxic and seemingly exceeding its known toxicity in the unresectable stage IV setting^{10, 14,22}. The first two, randomized arms of this trial-evaluated both single agent nivolumab and the combination of ipilimumab 3mg/kg and nivolumab 1mg/kg. 27% of patients treated with ipilimumab 3mg/kg and nivolumab 1mg/kg required surgical delays of 1-10 weeks due to need for steroids and prolonged steroid taper¹⁴. With no grade 3/4 IRAEs observed in the neoadjuvant setting and no confirmed toxicity related surgical delays, the combination of nivolumab and

relatlimab now provides a novel and alternative regimen that is highly effective with manageable toxicities in the neoadjuvant setting.

F. Improvements: Clarification of the following would be helpful:

1) Were any surgical quality control assessments done in the resection specimens? Current INMC recommendations are for full therapeutic dissection unless a protocol specifies otherwise.

Thank you for raising this important point. All patients underwent full therapeutic lymph node dissection with no “index nodal dissection” as in keeping with the recommendations of the International Neoadjuvant Melanoma Consortium.

2) Two “local” recurrences are noted. Were these local to the metastatic site or the primary? These are noted to have occurred 3 and 14 months after completing adjuvant therapy. Could the authors clarify the temporal relationship to surgery (i.e. did those patients get 10 doses of adjuvant therapy?)

Thank you for requesting information on this important area of clarification. There were two patients with pNR who experienced local recurrence events in soft tissues adjacent to prior surgical site. Each of these patients completed the full 10 doses of adjuvant therapy and had recurrence events 3 and 14 months, respectively, after completing the planned entire course of adjuvant therapy. This has been clarified in the Clinical Activity section on page 10 and lines 198-200.

3) The Menzies meta-analysis suggests overall and melanoma-specific survival in responders to neoadjuvant immunotherapy (there PD-1 monotherapy or ipi/nivo) are close to 100%. Although the numbers are small in the current study, 1 patient (8%) who had a pCR has died of metastatic melanoma, even with only the current early follow up. This raises the question of whether pathologic response (which does seem higher with this combination) is an unreliable surrogate endpoint to reflect OS or MSS benefit. Do the authors feel this argues for caution in relying too heavily on pCR to make decisions regarding regulatory approval or treatment preferences?

We appreciate the referee posing this important question. We respect the referee's call for caution. Without large multicenter, randomized trials, confirming the use of pCR as a surrogate endpoint for survival outcomes is challenging. Additionally, it is challenging to find an absolute predictor of outcomes for any treatment setting. However, as was shown with the Menzies analysis, insights may be gained from pooling studies to inform larger scale approaches. In the Menzies pooled analysis, patients treated with neoadjuvant immunotherapy who achieve any pathologic response (defined as pCR, near pCR and pPR) seem to derive better survival outcomes compared to those with pNR. Conversely, we have data to suggest that pCR alone is the best surrogate marker for patients treated with traditional cytotoxic chemotherapy in various solid tumors (breast, esophageal, gastric, rectal cancers) or BRAF/MEK targeted therapy in melanoma. Thus, the answer on whether pCR is the best surrogate marker for regulatory approval or treatment preferences may depend on the type of neoadjuvant therapy and mechanism of action being evaluated. This was addressed in the Discussion section on pages 18-19, lines 280-289.

G. References: No concerns

Thank you for this feedback.

H. Clarity: the manuscript is well written

Thank you for this feedback.

Referee #2 (Remarks to the Author):

This article reports on a single arm clinical trial testing the ability of a combination of anti-PD-1 and anti-LAG3 antibody therapy to induce pathologic responses in patients with surgically resectable melanoma.

Main comments:

What was the statistical design to result in 30 subjects being accrued, what was the target pathological response and the power of this trial to test the study hypothesis?

We thank the referee for this important question. The primary endpoint of this study was the determination of pathologic response rate to neoadjuvant treatment with nivolumab and relatlimab. We assumed a pathologic response rate of 30% for patients treated with this combination. Assuming this true pathologic response rate, the probability of at least 5 out of 30 patients experiencing a response is 0.97.

Were there different melanoma subtypes eligible? In particular, were there patients with acral, mucosal or desmoplastic melanoma included in this series?

We thank the referee for requesting this clarification. Yes, melanomas of all different subtypes were eligible for enrollment provided they had resectable, clinical stage III or oligometastatic stage IV disease. While all subtypes were allowed, only one patient with acral subtype was enrolled. There were no uveal, mucosal or desmoplastic melanoma patients enrolled on the study. This was clarified in the Methods section, Page 22, lines 474-476.

In the Abstract, the authors compare the relapse-free survival of patients with a biopsy with pathological response or no pathological response, and provide a p value. However, this p value is a result of non-randomized groups, and is prone to confounding effects of variables other than response to therapy.

Thank you for this insight. We agree, in this small, non-randomized study, there are potentially many confounding issues. However, the reporting of outcomes, particularly recurrence-free survival, following complete surgical resection has been reported in other neoadjuvant immunotherapy studies of similar size. Indeed, in the pooled analyses by Menzies et al, reporting outcomes based on any pathologic response to neoadjuvant ICB in patients with surgically resectable melanoma demonstrates favorable outcomes for patients. In addition, the comparison of responders vs non-responders within our study is appropriate as we try to identify clinical and biomarker features that could predict outcomes. Those comparisons have been performed with appropriate statistical assumptions and generated p-values that we believe we are bound to present to the readership of this journal. Please refer to our response to reviewer #1 for further details regarding statistical assumptions.

The Introduction is unnecessarily too long to put this study in context and justify its conduct.

Thank you so much for this critical feedback. We have reduced the length of introductory section and removed repetitive information.

There were 30 patients who initiated neoadjuvant treatment, and this should be the denominator for all efficacy and safety assessments. However, it seems like the pathological response rates are based on 29 patients, omitting a patient who progressed on therapy.

We thank the reviewer for this feedback. We initially reported the pathologic response based on the number of patients who went to surgery and thus had an assessment for pathologic response. However, we appreciate and agree with the referee's point and have corrected the clinical response rates in the abstract and throughout the manuscript to reflect the denominator of total treated patients, which now includes the one patient who did not proceed to surgery due to progression to metastatic disease during neoadjuvant therapy. However, in Fig 3 where we present results relating directly to pathological assessment of the resection specimen including comparisons of radiographic and pathologic response, we felt that to preserve the accuracy of the observations only patients who have undergone surgery are evaluable for this outcome (for instance, the patient with metastatic disease could have conceivably had a pCR in the locoregional area). We therefore used the denominator of 29 patients for those data and figures.

The first paragraph of the Discussion is an unnecessary numerical repetition of the results that are just immediately before presented in the Results section. It would be desirable to shorten this paragraph to at most a couple of sentences describing the top line results without repeating all the numbers. As with the Introduction, the Discussion seems too long for the data being presented.

We appreciate this feedback and have revised the first paragraph of the discussion to remove the repetition of the results. Additionally, we shortened the entire first paragraph and edited the entire Discussion section to remove unnecessary information.

Minor comments:

One subject who consented to participate was not enrolled due to "insurance denial". However, the clinical trial provided the study drugs. What insurance denial led to inability to participate?

Thank you for this question. While the sponsor of the clinical trial did provide the study drugs free of cost, there are some insurance plans that do not permit participation in clinical trials despite free drugs. As this trial did require insurance to cover the costs of standard of care assessments such as physical exam co-pays, labs and scans, insurance providers have unfortunately opted to prohibit clinical trial enrollment.

One subject is reported to having passed away 14 months after surgery with an unconfirmed disease progression in the brain (lines 208-209). Is this "unconfirmed" due to an unclear lesion, or because there were not two serial scans, or because there was no biopsy? If the patient develop brain metastases by radiologic assessment and died of progressive disease, the point of being "unconfirmed" is not relevant.

Thank you for this important clarification. We describe unconfirmed disease progression in the brain as the study team was informed by the patient that he experienced disease progression in the brain however he withdrew consent for further clinical trial participation and we were never able to confirm presence of brain metastases by any available medical records. We do have evidence of a documented date of death but no definitive cause of death.

The "NR patients" and "R patients" in lines 230-232 seem to refer to patients whose pathological assessment had a response or response to therapy. It would be desirable to not make the patients an adjective of their tumor response.

We appreciate this feedback and have re-written the corresponding sentences.

Referee #3 (Remarks to the Author):

Amaria et al. reported a study with relatlimab and nivolumab as neoadjuvant for patients with resectable stage III or oligometastatic stage IV melanoma. Patients received 2 neoadjuvant doses followed by surgery, and then 10 doses of adjuvant combination therapy.

The use of checkpoint inhibitors combinations is an important strategy to overcome resistance and improve survival. In particular, the anti-LAG-3 relatlimab in combination with nivolumab showed positive results in a recent phase III study in metastatic melanoma patients (Tawbi et al NEJM 2022) with a subsequent approval from FDA. The study from Amaria et al. is the first which evaluated an anti-LAG-3 in combination with anti-PD-1 as neoadjuvant therapy. The data looks certainly interesting in comparison with all the data already published in this field. Thirty patients were enrolled and the combination resulted in 59% pCR rate and 73% overall pathological response rate. The safety was consistent with all the previous reports from this new combination (Ascierto et al ESMO 2017; Tawbi et al NEJM 2022) showed and incidence of AEs G3-4 slightly higher compared to anti-PD-1 monotherapy. Excellent results if we consider that with the combination ipilimumab/nivolumab the pCR rate was 50-60% in previous reports with a surely higher incidence of side effects.

We thank the reviewer for this thorough summary of our manuscript in the context of recent clinical trial data using the combination of relatlimab and nivolumab.

However, if I certainly agree about the more favorable safety profile, in term of efficacy at the moment this data should be considered preliminary results since come from a small cohort of patients. Moreover, if the pCR is the direct effect of the neoadjuvant treatment, the RFS may be affected by the subsequent adjuvant treatment. Indeed, the subsequent adjuvant therapy with the combination could be a confounding factor to define the efficacy of the neoadjuvant strategy. This aspect should be discussed in the manuscript.

We appreciate the referee's thoughtful feedback. We do agree these results are preliminary due to the small sample size and relatively short follow up interval. This was more explicitly laid out in the Discussion section on pages 15-16, lines 323-328. Regarding the point that RFS may be affected by subsequent adjuvant therapy, we are in complete agreement with this statement and indeed the effect of adjuvant therapy remains a matter of debate in neoadjuvant trials as discussed in the Discussion section on page 14, lines 275-278.

In the manuscript it should also be discussed the risk that results could be affected by patients selection. The fact that the majority of patients were BRAF wild type and the median size of the lesions was 26 mm might indicate a selection bias. If it's true that the Authors described an absence of correlation between the tumor lesions size and response, the small number of patients enrolled doesn't allow to make any conclusion. These are surely interesting data but still preliminary.

We thank the referee for these observations. We do agree that the generalizability of these results to patients with BRAF mutated melanoma may be limited due to the small proportion of these patients included in the trial and have made note of this in the Discussion section, page 15, lines 298-304. Regarding the comment on modest tumor size causing potential selection bias, we did specifically analyze response in patients with larger burden disease and did not identify less robust responses in this group. This can be found in the Discussion section on page 15, lines 293-295.

As regard to the correlative studies, flow cytometry and mass cytometry (CyTOF) analyzes were performed on tumor specimens as well as PBMCs. Biomarker analysis showed a high frequency of CD45+ in the pretreatment responders' samples compared to pretreatment samples of non-responder

and an increase of effector CD8+ T cell subset and a memory CD4+ T cell subset was seen in responding versus non responding.

Another interesting evidence is the decrease of M2 macrophages in patients with pathologic response in TME over the course of neoadjuvant therapy. Unfortunately, it is not described clearly the methodology of this finding. M2 detection is usually evaluated with CD68 +, CD206+, CD163+ and for some studies also PDL1+. In this work, the identification of M2 occurs with several different receptors which could select a larger population of cells not specifically M2. Authors should better discuss this aspect.

We thank the reviewer who has raised interesting questions. In Extended Figure 3, we reported a population of “M2-Like” cells which were CD68+ HLA-DR+ CD14+ VISTA+ CD163+ CD45RO+ PD-L1+. The phenotyping and identification of M2 populations are matters of ongoing discussion among immunologists, with several subsets being reported. As the reviewer has correctly stated, CD68 and CD163 have indeed been reported as markers of M2 macrophages (Hu et al., <https://www.ncbi.nlm.nih.gov/pmc/articles/PMC5400603/> and Hensler et al. <https://jitc.bmj.com/content/8/2/e000979>). As such, we have used these markers to identify the population as “M2-Like”. We did not include CD206 in this CyTOF panel because of the evidence that some M1 macrophages do express CD206, *albeit* at a lower level compared to M2 (<https://currentprotocols.onlinelibrary.wiley.com/doi/full/10.1002/cpsc.108>). The expression of VISTA, HLA-DR, and CD45RO does suggest that there are different subsets of M2 macrophages (Gao et al., <https://www.ncbi.nlm.nih.gov/pmc/articles/PMC5466900/>, and <https://www.ncbi.nlm.nih.gov/pmc/articles/PMC2759556>). We acknowledge that without further functional and cytokine assays, we cannot identify these cells with more specificity, and have thus chosen to describe them as “M2-Like”.

Finally, it would be interesting to get data from a complete mutational analysis by NGS on both pretreatment and posttreatment tumor tissue.

We appreciate this thoughtful comment from the referee. We agree complete NGS would be interesting on these patients but, unfortunately, we are limited by lack of available DNA for analysis. Future studies using RNA sequencing are planned for samples with available genetic material.

Referee #4 (Remarks to the Author):

This is a straightforward single arm trial, and another good attempt to find effective neoadjuvant therapies with modern drugs.

We thank the referee for this assessment.

1. The design parameters are missing from the paper: what was the target pCR rate and what background rate was assumed in the sample size calculation? These are essential in order to see whether the target was achieved. Also, an appropriate confidence interval needs to be provided for the pCR rate.

We thank the referee for this important question. The primary endpoint of this study was the determination of pathologic response rate to neoadjuvant treatment with nivolumab and relatlimab. As this is the first evaluation of this combination in the neoadjuvant setting, we assumed a pathologic response rate of 30% for patients treated with this combination – the pathologic response rate observed from single agent nivolumab. Assuming this true pathologic response rate, the probability of at least 5 out of 30 patients experiencing a response is 0.97.

2. There needs to be Kaplan-Meier curves for EFS, RFS and OS (in the appendix) for all patients (in addition to showing curves split by response). In the Discussion, say how the observed results for these outcomes compare with those seen in patients in routine practice?

We thank the referee for this feedback and question. We have added recurrence free survival (RFS) for all patients who went to surgery to Extended data figure 1A and Extended data figure 1B demonstrates the RFS by pCR vs non pCR. Additionally, we have included in Extended data figure 4, the event free survival (EFS) curve for all patients. Finally, the Overall Survival (OS) curve for all patients is shown on Figure 3d. We have not included an OS figure separated by pathologic response as not all patients made it to surgery for pathology assessment.

3. The definitions of RFS and EFS were unclear. It is usual that death from any cause is counted as an event; is this correct in this study? The text states that “patients who died without experiencing progression/recurrence were censored”, which implies that someone who died from melanoma (without progression) would be censored – but this should not be the case, they should be counted as an event (there are several patients like this in clinical trials, it just means they died before progression could be clinically ascertained). Also, for OS, patients who have not died should be censored at the date of last follow up (unclear what ‘last known vital status’ actually means).

We appreciate the referee highlighting the opportunity to clarify these analyses. The event-free survival (EFS), as described on page 25, lines 519-520, is computed as events from the time of treatment start until their discontinuation due to progression or death. There was one patient who discontinued due to progression prior to surgery and thus event is captured prior to surgery. As such, we have not included an EFS curve by pathologic response due to the fact that not all patients were evaluable for pathologic response. We appreciate the reviewer pointing out the text “patients who died without experiencing progression/recurrence were censored”. This is, as pointed out, incorrect and has been removed from the text. These patients were counted as events in the EFS analysis. An EFS curve for all patients has been included in Extended data figure 4.

The recurrence-free survival (RFS) is defined as any recurrence or death following surgery (defined on page 25, lines 518-519). The patient who progressed prior to surgery is not included

in this RFS analysis. We have slightly rearranged the order of reporting in the manuscript to assist in mitigating confusion between the EFS and RFS analyses. Additionally, as mentioned above, we have included RFS for all patients and also RFS by pCR and non pCR – Supplemental Figure 1A and 1B.

Finally, OS was calculated as described – with the last known date of follow up and alive or date of death.

4. Could the high pCR rate be partly influenced by all patients coming from 2 highly expert cancer centres, nearly all patients were ECOG 0, or 60% stage IIIb?

We thank the referee for this question. We know that real-world data tends to be not as favorable as clinical trial data and this is appreciated in every aspect of medical research. However, in this specific situation, we would like to point out that the vast majority of clinical stage III melanoma patients truly have disease characteristics seen in this trial—ECOG 0, normal LDH and modest burden disease. Additionally, while patients were enrolled in large-volume academic centers with experience in neoadjuvant trials, it is unclear how this would influence the pCR rate. The pathologists at these centers were instrumental in defining the pathologic response criteria after neoadjuvant therapy in melanoma and thus we would argue that the pCR rate is accurate and that the guidelines are clear enough that the pCR rate would be found to be reproducible if the study were conducted at other sites. Finally, it should be noted that at this stage in neoadjuvant therapy of melanoma, most of the reported studies have indeed been conducted at large highly expert cancer centers including the NKI, the MIA, the University of Pennsylvania, Moffitt Cancer Center, and the University of Pittsburgh among others.

Reviewer Reports on the First Revision:

Referees' comments:

Referee #1 (Remarks to the Author):

Thank you to the authors for your thoughtful consideration of reviewer critiques. My only remaining minor suggestion is to temper the use of “significant” when describing comparisons in the correlative studies portion of the manuscript. Given the number of different tests that were done, a p-value of 0.05 is probably not a great indicator of significance (at least statistically). You might simply report the differences with p-values and let the reader draw his/her own conclusions about significance.

Referee #2 (Remarks to the Author):

The authors provide a marginally improved article after the consistent comments from the initial review.

The key point is if this clinical trial had a prospective statistical design that was included in the protocol and resulted in a power calculation to enroll 30 eligible patients. The response to the reviewers refers to wording that seems to be some kind of a prospective design. But a prospective clinical trial design was and is not described in the Methods or Results. It would be useful for the reviewers to have access to the statistical design section of the clinical trial protocol as it was when the trial was being conducted.

Others minor points continue to need to be addressed, such as the unnecessary length of the Introduction and Discussion, mostly going over points of minor interest to the majority of the readers that relates mostly to the investigator's prior neoadjuvant trials.

For the “Consort” diagram in Figure 2, it would be desirable that the authors reviewed other articles with such diagrams and improved the figure.

Referee #3 (Remarks to the Author):

The Authors addressed and satisfied all my comments and those made by the other reviewers.

Referee #4 (Remarks to the Author):

2 reviewers specifically asked for the sample size justification used in the design to go into the text (Methods). Please add this, making sure it is the one in the trial protocol.

A 95%CI was requested for the main endpoint pCR but not given in the revised paper. For 17/30, the exact 95%CI is 37-74%; and the authors can state that even the lower limit (37%) exceeds their minimum target of 30%.

When defining EFS/RFS just say: "EFS time was computed from start of treatment to date of progression/recurrence or death from any cause, whichever occurred first."

All of my other comments seem to have been addressed.

Author Rebuttals to First Revision:

Referee #1 (Remarks to the Author):

Thank you to the authors for your thoughtful consideration of reviewer critiques. My only remaining minor suggestion is to temper the use of “significant” when describing comparisons in the correlative studies portion of the manuscript. Given the number of different tests that were done, a p-value of 0.05 is probably not a great indicator of significance (at least statistically). You might simply report the differences with p-values and let the reader draw his/her own conclusions about significance.

We thank the referee for their review of our revised manuscript and for this thoughtful point and recommendation. We have removed the word “significant” from the Correlative Studies portion of the manuscript.

Referee #2 (Remarks to the Author):

The authors provide a marginally improved article after the consistent comments from the initial review.

The key point is if this clinical trial had a prospective statistical design that was included in the protocol and resulted in a power calculation to enroll 30 eligible patients. The response to the reviewers refers to wording that seems to be some kind of a prospective design. But a prospective clinical trial design was and is not described in the Methods or Results. It would be useful for the reviewers to have access to the statistical design section of the clinical trial protocol as it was when the trial was being conducted.

We thank the referee for their review of our revised manuscript and for the recommendation to clarify the design of this prospective clinical trial. We have added further clarification in the manuscript under the Study Design and Methods sections. We are happy to provide the entire clinical trial protocol for review.

Others minor points continue to need to be addressed, such as the unnecessary length of the Introduction and Discussion, mostly going over points of minor interest to the majority of the readers that relates mostly to the investigator’s prior neoadjuvant trials.

We thank the referee for this feedback. We have reduced the length of the Introduction. We critically reviewed the Discussion for areas to cut but want to ensure we have kept the main points in order to appropriately contextualize the results within the field of neoadjuvant melanoma treatment.

For the “Consort” diagram in Figure 2, it would be desirable that the authors reviewed other articles with such diagrams and improved the figure.

We thank the referee for this comment and have altered the Consort Diagram/Figure 2 based on your feedback.

Referee #3 (Remarks to the Author):

The Authors addressed and satisfied all my comments and those made by the other reviewers.

We thank the referee for their review of our revised manuscript.

Referee #4 (Remarks to the Author):

2 reviewers specifically asked for the sample size justification used in the design to go into the text (Methods). Please add this, making sure it is the one in the trial protocol.

We thank the referee for their review of our revised manuscript and for the recommendation to clarify the design of this prospective clinical trial. We have added further clarification in the manuscript under the Study Design and Methods sections.

A 95%CI was requested for the main endpoint pCR but not given in the revised paper. For 17/30, the exact 95%CI is 37-74%; and the authors can state that even the lower limit (37%) exceeds their minimum target of 30%.

We thank the referee for this thoughtful recommendation and have amended the manuscript accordingly in both the Clinical Activity and Discussion portion of the manuscript.

When defining EFS/RFS just say: "EFS time was computed from start of treatment to date of progression/recurrence or death from any cause, whichever occurred first."

We thank the referee for this suggestion and, again, for their original point in the first review. We feel it is important to distinguish RFS from EFS as they are two distinct evaluations with two different starting points. In the Statistical Analyses portion of the manuscript we have described these distinctions as we report outcomes utilizing both methods.

All of my other comments seem to have been addressed.

We thank the referee, again, for their review of our revised manuscript.

Reviewer Reports on the Second Revision:

Referees' comments:

Referee #1 (Remarks to the Author):

No additional comments. The authors have addressed the prior review suggestions/critiques.

Referee #2 (Remarks to the Author):

The authors have correctly addressed the comments from the reviewers.

Referee #4 (Remarks to the Author):

Comments have been addressed now.

Author Rebuttals to Second Revision:

Referees' comments:

Referee #1 (Remarks to the Author):

No additional comments. The authors have addressed the prior review suggestions/critiques.

We thank the Referee for the constructive comments and feedback during this review process.

Referee #2 (Remarks to the Author):

The authors have correctly addressed the comments from the reviewers.

We thank the Referee for the constructive comments and feedback during this review process.

Referee #4 (Remarks to the Author):

Comments have been addressed now.

We thank the Referee for the constructive comments and feedback during this review process.